# Three Dimensional Topological Field Theories and Nahm Sum Formulas

**Dongmin Gang,**[1] **Heeyeon Kim,**[2] **Byoungyoon Park,**[1] **and Spencer Stubbs**[3]

[1]*Department of Physics and Astronomy & Center for Theoretical Physics, Seoul National University, 1 Gwanak-ro, Seoul 08826, Korea*

[2]*Department of Physics, Korea Advanced Institute of Science and Technology,Daejeon 34141, Republic of Korea*

[3]*NHETC and Department of Physics and Astronomy, Rutgers University, 126 Frelinghuysen Rd., Piscataway NJ 08855, USA*

ABSTRACT: It is known that a large class of characters of 2d conformal field theories (CFTs) can be written in the form of a Nahm sum. In [1], D. Zagier identified a list of Nahm sum expressions that are modular functions under a congruence subgroup of $SL(2,\mathbb{Z})$ and can be thought of as candidates for characters of rational CFTs. Motivated by the observation that the same formulas appear as the half-indices of certain 3d $\mathcal{N}=2$ supersymmetric gauge theories, we perform a general search over low-rank 3d $\mathcal{N}=2$ abelian Chern-Simons matter theories which either flow to unitary TFTs or $\mathcal{N}=4$ rank-zero SCFTs in the infrared. These are exceptional classes of 3d theories, which are expected to support rational and $C_2$-cofinite chiral algebras on their boundary. We compare and contrast our results with Zagier's and comment on a possible generalization of Nahm's conjecture.

## 1   Introduction

The classification of two-dimensional rational conformal field theories (RCFT) has been extensively studied in both physics and mathematics, particularly following the seminal work by Mathur, Mukhi, and Sen [2]. The key feature of RCFTs that plays a central role in the classification program is the modularity of the characters, defined as

$$\chi_{M_i}(q) = q^{-c/24} \operatorname{Tr}_{M_i} q^{L_0} \ , \tag{1.1}$$

where $\{M_1, \cdots, M_d\}$ are simple modules of an underlying rational and $C_2$-cofinite vertex operator algebra (VOA) [3]. These characters $\{\chi_{M_1}, \cdots \chi_{M_d}\}$ transform as a vector-valued modular function under $SL(2, \mathbb{Z})$,

$$\chi_{M_j}(\tau') = R_{ij}\chi_{M_i}(\tau) \ , \qquad \tau' = \frac{a\tau + b}{c\tau + d} \ , \quad \begin{pmatrix} a & b \\ c & d \end{pmatrix} \in PSL(2, \mathbb{Z}) \ , \tag{1.2}$$

where $R_{ij}$ is a representation of $SL(2, \mathbb{Z})$. It is known that each of the characters $\chi_{M_i}(q)$ transforms as a modular function of weight zero under some congruence subgroup $\Gamma(n)$. Although the converse is not expected to be true in general, such modular functions are natural candidates for the characters of a rational vertex operator algebra offering an interesting approach for the classification of 2d RCFTs [2, 4–8].

Among various strategies, we focus on a series of works [1, 9–12], which highlighted the fact that a large class of 2d VOA characters can be represented as a *Nahm sum formula*[1].

$$\chi_{(A,B,C)}(q) = \sum_{m \in \mathbb{N}^r} \frac{q^{\frac{1}{2}m^t A m + B^t m + C}}{(q)_{m_1} \cdots (q)_{m_r}} \ , \tag{1.3}$$

where $A$ is a positive semi-definite $r \times r$ matrix, $B$ is a length $r$ vector, and $C$ is a real number. The most familiar example is given by the vacuum character of the Virasoro minimal model $M(2,5)$,

$$\chi_0(q) = q^{11/60} \prod_{n=2,3 (\mathrm{mod}\ 5)} \frac{1}{(1-q^n)} = \sum_{n=0}^{\infty} \frac{q^{n^2+n+11/60}}{(q)_n} \ . \tag{1.4}$$

Other examples of characters with Nahm sum representations or generalizations thereof include those of some Virasoro minimal models, super-Virasoro minimal models, lattice VOAs, and log-VOAs. See [13–16] for a partial list of related works.

A natural question is the following: for which matrices $A$ do there exist $B$ and $C$ such that (1.3) becomes a modular function? W. Nahm attempted to answer this question by developing a conjecture which relates modular functions and torsion elements in the Bloch group. Utilizing this conjecture, Nahm, and separately D. Zagier, found numerous examples of triplets $(A, B, C)$ for which (1.3) is modular [1, 12].

This paper is motivated by an observation that the Nahm sum formula (1.3) also appears naturally as a half-index of 3d $\mathcal{N} = 2$ $U(1)_K^r$ Chern-Simons matter (CSM) theories of type

$$U(1)_K^r \ + \ r \text{ chiral multiplets} \tag{1.5}$$

where each of the $U(1)$ factors has precisely one charged chiral multiplet of charge $1$[2]. In this context, the matrix $A$ is identified with the mixed Chern-Simons level matrix $K$. As discussed extensively in recent literature [24–28], the boundary vertex operator algebras of 3d supersymmetric gauge theories are generically non-rational. However, there are two important classes of exceptions:

(i) The theory flows to a unitary TFT.

(ii) The theory flows to a rank-zero SCFT with supersymmetry enhancement to $\mathcal{N} = 4$.

If (ii) occurs, one can perform the full topological A/B-twist to obtain a pair of non-unitary semi-simple TFTs which admit boundary conditions that support a 2d rational VOA. This point of view has been useful in the recent discovery of novel bulk-boundary relations involving a large class of non-unitary, rational VOAs [19–23, 29–35]; motivating us to perform a general search for $\mathcal{N} = 2$ abelian CSM theories of type (1.5) that satisfy (i) or (ii). In order to flow to a rank-zero SCFT or a unitary TFT, this description is generally subject to a superpotential deformation involving monopole operators. We search for integer matrices $K$ for which there is such a superpotential deformation leading to the desired infrared (IR) behavior. The search is performed over positive definite $K$ for $r = 1, 2, 3$ with entries ranging from -17 to 17. While the bounds for $K_{ij}$ are chosen primarily for technical reasons, we expect that gauge theories with large values of $|K_{ij}|$ are unlikely to flow to a rank-zero SCFT. This is because the quantum dimensions of monopole operators, which are roughly proportional to the CS level, become too large and render the superpotential deformation irrelevant.

The results are summarized in Section 4. We find 27 distinct examples and an infinite family of $K$ that are expected to flow to rank-zero SCFTs. It is not necessary for each IR theory to be

---

[1]$\mathbb{N}$ is a set of non-negative integers including 0.
[2]See recent works [17–23], which focuses on a similar observation.

distinct, as an abelian Chern-Simons matter theory often has a large duality orbit. We perform extensive tests of IR dualities among the candidate theories by computing various supersymmetric observables, and find that they can be organized into 8 duality classes. We do not perform an exhaustive search for theories that flow to unitary TFTs, but are able to identify several infinite families.

We compare the result with the list in [1], and find some examples of modular functions that do not appear in *loc.cit.*. This is due to the fact that a slightly modified version of the Nahm sum formula is more natural from the physical point of view, leading to a small generalization of Nahm's conjecture. We also find that there are several examples which appear in *loc.cit.* but do not show up in our search, and explain why this is expected.

This paper is organized as follows. In Section 2, we give a short review of Nahm's conjecture and reformulate it in the point of view of 3d supersymmetric quantum field theory. In Section 3, we review the general aspects of 3d abelian CSM theories; listing the necessary conditions for these theories to flow to rank-zero SCFTs in the IR. In Section 4, we summarize the results of our search and list the candidate abelian CSM theories that flow to rank-zero SCFTs or unitary TFTs. In Section 5, we collect a partial list of open questions. In Appendix A, we summarize our conventions for supersymmetric partition function computations. In Appendix B, we collect useful formulas for various characters of rational VOAs.

## 2 Nahm sum formula and modular functions

Consider the system of equations

$$1 - x_i = \prod_{j=1}^{r} x_j^{A_{ij}} , \tag{2.1}$$

which is obtained by examining the asymptotic behavior of the Nahm sum formula (1.3) as $q \to 1$. Nahm's conjecture can be stated in a way that relates the solutions of (2.1) to the modularity of (1.3). This is achieved via two special functions known as the Rogers dilogarithm and the Bloch-Wigner function. The *Rogers dilogarithm* is defined on $0 < x < 1$ as

$$L(x) = \mathrm{Li}_2(x) + \frac{1}{2} \log(x) \log(1 - x). \tag{2.2}$$

This function satisfies $\lim_{x \to 0} L(x) = 0$ and $\lim_{x \to 1} L(x) = \frac{\pi^2}{6}$ which can be taken as definitions for $L(0)$ and $L(1)$ respectively. The function can then be extended to the rest of $\mathbb{R}$ as

$$L(x) = \begin{cases} 2L(1) - L(1/x) & \text{if } x > 1, \\ -L(x/(x-1)) & \text{if } x < 0. \end{cases} \tag{2.3}$$

The *Bloch-Wigner function* is defined as

$$D(z) = \mathrm{Im}(\mathrm{Li}_2(z)) + \arg(1 - z)\log|z|. \tag{2.4}$$

This function is continuous for all $z \in \mathbb{C}$ and satisfies $D(z) = 0$ for $z \in \mathbb{R}$. For $A$ positive definite and symmetric with entries in $\mathbb{Q}$, (2.1) has exactly one solution with all real algebraic entries between 0 and 1. We denote the solutions to (2.1) as $X_i^{(a)}$ with $a$ indexing the solutions and beginning at $X_i^{(0)}$, which is defined to be the special solution with $0 < X_i^{(0)} < 1$. With this information Nahm's conjecture can be stated as follows[3].

---

[3]This is not the original form of Nahm's conjecture as discussed in [1, 12] for which a counterexample was found in [36]. The conjecture given here is equivalent to a weakened which is discussed in [37].

**Nahm's conjecture.** Let A be a positive definite symmetric $r \times r$ matrix with rational entries and let $X^A$ be the corresponding set of solutions to (2.1). Given the following statements:

(i) For all solutions in $X_i^{(a)} \in X^A$, $\sum_i D(X_i^{(a)}) = 0$.

(ii) For the special solution $X_i^{(0)}$, $\frac{1}{L(1)} \sum_i L(X_i^{(0)}) \in \mathbb{Q}$.

(iii) There exist $B \in \mathbb{Q}^r$ and $C \in \mathbb{Q}$ such that $\chi_{A,B,C}(q)$ is a modular function.

Then $(i) \implies (iii)$ and $(iii) \implies (ii)$.

Our work relates to Nahm's conjecture in several ways. In section 4, we perform a generic search for matrices $K_{ab}$ that define a class of $\mathcal{N} = 2$ Chern-Simons matter theories which flow either to rank-0 $\mathcal{N} = 4$ SCFTs or unitary TFTs. As will be discussed in the next section, the partition function of these theories on $D^2 \times S^1$ with a specific choice of the boundary conditions, an object often called the half-index, can be schematically written in the form

$$I_{\text{half}} = \sum_{m \in \mathbb{N}^r} (-1)^{\alpha^t m} \frac{q^{\frac{1}{2} m^t K m + \beta^t m + \gamma}}{(q)_{m_1} \cdots (q)_{m_r}} . \tag{2.5}$$

This is nearly the same as (1.3), the difference being an additional sign factor in the summand. This sign is related to the choice of spin structure along the non-contractable cycle of the boundary torus. Furthermore, the partition functions of 3d theories on Seifert manifolds can be written as a sum over solutions to the *Bethe equations*

$$1 - x_a = \zeta_a \prod_{b=1}^{r} x_b^{K_{ab}}, \tag{2.6}$$

which can be obtained by asymptotic analysis of (2.5). These equations are the same as (2.1) up to an added phase factor $\zeta_a$. Moreover, as described in Appendix A, the data of a UV abelian Chern-Simons matter theory with the desired IR properties can be used to determine the modular $S$ and $T$ matrices of a related semi-simple TFT. One can show that when the TFT is non-spin the matrix $T$ takes the form[4]

$$T_{\alpha\beta} = \delta_{\alpha\beta} \exp\left[ \frac{1}{2\pi i} \sum_i L(X_i^{(\alpha)}) \right]$$

and that the modulus is

$$|T_{\alpha\alpha}| = \exp\left[ \frac{1}{2\pi} \left( \sum_i D(X_i^{(\alpha)}) \right) \right]$$

with $\alpha$ labeling solutions to equations of the form (2.1) obtained from the UV theory. Using these facts, we can develop a physical interpretation of $(i)$ and $(ii)$ in Nahm's conjecture:

$$(i) \implies |T_{\alpha\alpha}| = 1 \text{ for all } X_i^{(\alpha)} \in X^A,$$
$$(ii) \implies \arg(T_{\alpha\alpha}) \in \pi\mathbb{Q} \text{ for some solution } X_i^{(\alpha)} \in X^A.$$

These are both necessary conditions on the modular data of a semi-simple TFT. In fact, in physics we expect that $\arg((T_{\alpha\alpha})) \in \pi\mathbb{Q}$ for all solutions to the equation (2.1).

The connection between Nahm's conjecture and our work gives us a body of previous results in the math literature with which to compare and contrast. In [1], Zagier performed a general search over low-rank matrices ($r = 1, 2, 3$) satisfying $(i)$. He found a total of four infinite families as well

---

[4]Up to overall normalization by a unit norm complex number.

| | |
|---|---|
| $r=1$ | $2 \quad 1 \quad 1/2$ |
| $r=2$ | $\begin{pmatrix} \alpha & 1-\alpha \\ 1-\alpha & \alpha \end{pmatrix}$ $\begin{pmatrix} 2 & 1 \\ 1 & 1 \end{pmatrix}$ $\begin{pmatrix} 4 & 1 \\ 1 & 1 \end{pmatrix}$ $\begin{pmatrix} 4 & 2 \\ 2 & 2 \end{pmatrix}$ $\begin{pmatrix} 2 & 1 \\ 1 & 3/2 \end{pmatrix}$ $\begin{pmatrix} 4/3 & 2/3 \\ 2/3 & 4/3 \end{pmatrix}$ |
| $r=3$ | $\begin{pmatrix} 2 & 1 & -1 \\ 1 & 1 & 0 \\ -1 & 0 & 1 \end{pmatrix}$ $\begin{pmatrix} 2 & 1 & 1 \\ 1 & 1 & 0 \\ 1 & 0 & 1 \end{pmatrix}$ $\begin{pmatrix} 3 & 2 & 1 \\ 2 & 2 & 1 \\ 1 & 1 & 1 \end{pmatrix}$ $\begin{pmatrix} 2 & 1 & 1 \\ 1 & 2 & 0 \\ 1 & 0 & 2 \end{pmatrix}$ $\begin{pmatrix} 4 & 2 & 1 \\ 2 & 2 & 0 \\ 1 & 0 & 1 \end{pmatrix}$ $\begin{pmatrix} 6 & 4 & 2 \\ 4 & 4 & 2 \\ 2 & 2 & 2 \end{pmatrix}$ $\begin{pmatrix} 4 & 2 & 2 \\ 2 & 2 & 1 \\ 2 & 1 & 2 \end{pmatrix}$ $\begin{pmatrix} 4 & 2 & -1 \\ 2 & 2 & -1 \\ -1 & -1 & 1 \end{pmatrix}$ $\begin{pmatrix} 8 & 4 & 1 \\ 4 & 3 & 0 \\ 1 & 0 & 1 \end{pmatrix}$ $\begin{pmatrix} \alpha h^2+1 & \alpha h & -\alpha h \\ \alpha h & \alpha & 1-\alpha \\ -\alpha h & 1-\alpha & \alpha \end{pmatrix}$ $\begin{pmatrix} \alpha h^2+2 & \alpha h & -\alpha h \\ \alpha h & \alpha & 1-\alpha \\ -\alpha h & 1-\alpha & \alpha \end{pmatrix}$ $\begin{pmatrix} \alpha h^2+1/2 & \alpha h & -\alpha h \\ \alpha h & \alpha & 1-\alpha \\ -\alpha h & 1-\alpha & \alpha \end{pmatrix}$ |

**Table 1**. The matrices found in [1]. For each matrix listed, its inverse is also a valid result. Several infinite families are listed with $\alpha \in \mathbb{Q}$, $h \in \mathbb{Z}$.

as several sporadic cases all listed in Table 1. For each case in Table 1 there exists at least one pair $(B, C)$ such that $\chi_{(A,B,C)}(\tau)$ is modular. However, due to the implication structure in Nahm's conjecture, the matrices in Table 1 only classify a subset of the low-rank cases resulting in modular functions. Additionally, in [12] Nahm proposed a family of matrices categorized by diagrams of ADET type. Here, "ADE" stands for the Dynkin diagrams $A_r$, $D_r$ and $E_r$, while "T" is less familiar. The diagram $T_r$ is defined as an $A_{2r}$ Dynkin diagram folded in half. Its primary difference in this context is that the Cartan matrix associated with $T_r$ is the same as that of $A_r$ except for $C(T_r)_{rr} = 1$. Given a pair $(X, Y)$ of ADET type diagrams, Nahm proposed that

$$A(X,Y) = C(X) \otimes C(Y)^{-1} \tag{2.7}$$

would satisfy $(i)$. Solutions to (2.1) for specific families $(X, Y)$ were studied and determined analytically in [12, 38] and a proof that matrices of the form (2.7) satisfy $(i)$ was claimed in [39].

The previous work on Nahm's conjecture is best described as a bellwether rather than a precise target we would wish to replicate. For one, because we study Chern-Simons levels, all of our results are integer-valued matrices; a restriction that does not exist when approaching this problem from a purely mathematical point of view. On the other hand, when comparing our findings with those of Nahm and Zagier, we should expect agreement only when their $(A, B)$ are integral.[5] Further, the possible appearance of an additional phase in (2.6) indicates that we are working with a slight generalization of Nahm's conjecture, where $L(x)$ is modified by a linear term in $\log(x)$ (See Sec. A.4 for a discussion). Therefore, we should expect to discover new matrices beyond those originating from Nahm's conjecture. Indeed, this is what we find: any level matrix $K$ listed in Section 4 that is not also given in Table 1 either does not satisfy $(i)$ (and thus was ignored by Zagier) or does

[5]The integrality of $B$ arises from the physical interpretation of the linear term in the half index.

not satisfy the conditions of Nahm's conjecture due to the more general form of (2.5). For these novel cases in Section 4, we find that there are negative contributions in the character summation formula, i.e. (2.5) does not reduce to (1.3) and thus does not contradict the partial proof of Nahm's conjecture in [37].

## 3 A class of rank-zero theories

### 3.1 $\mathcal{N} = 2$ abelian Chern-Simons matter theories

We consider a class of $\mathcal{N} = 2$ abelian Chern-Simons matter theories, which are labeled by the Chern-Simons level matrix $K$ and a set of chiral operators $\{\mathcal{O}_I\}$

$$\mathcal{T}[K, \{\mathcal{O}_I\}] := \frac{(\mathcal{T}_\Delta)^r}{U(1)_K^r} \text{ with superpotential } \mathcal{W} = \sum_{I=1}^{N_\mathcal{O}} \mathcal{O}_I . \tag{3.1}$$

Here $\mathcal{T}_\Delta$ is a free theory of a single chiral field $\Phi$ with background Chern-Simons level $-\frac{1}{2}$ [40] and a Lagrangian density given by

$$\mathcal{L}_{\mathcal{T}_\Delta}(V) = \int d^4\theta \left( -\frac{1}{8\pi} \Sigma_V V + \Phi^\dagger e^V \Phi \right) . \tag{3.2}$$

Here $\Sigma_V$ is the linear superfield containing the field strength of the background vector multiplet $V$ coupled to the $U(1)$ flavor symmetry. The theory also has $U(1)$ R-symmetry. The mixed CS levels between the R-symmetry and the flavor symmetry are summarized in Table 2. The notation $/U(1)_K^r$

| $F(\Phi)$ | $R_*(\Phi)$ | $k_{FF}$ | $k_{FR_*}$ |
|-----------|-------------|----------|------------|
| 1 | 0 | $-\frac{1}{2}$ | $\frac{1}{2}$ |

**Table 2**. Charges ($F$ and $R_*$) of a chiral field $\Phi$ in the $\mathcal{T}_\Delta$ theory under the $U(1)$ flavor and $U(1)$ R-symmetry and mixed background CS levels ($k_{FF}$ and $k_{FR_*}$). The superconformal R-charge of the free chiral theory $\mathcal{T}_\Delta$ corresponds to $R_* + \frac{1}{2}F$.

denotes the supersymmetric gauging of the $U(1)^r$ flavor symmetry in the $(\mathcal{T}_\Delta)^r$ theory ($r$ copies of $\mathcal{T}_\Delta$) with mixed Chern-Simons level $K$. The gauge charge $Q_{ij}$ of the $i$-th chiral field $\Phi_i$ under the $j$-th $U(1)$ gauge symmetry is chosen to be $\delta_{ij}$. If we denote $\{\mathcal{V}_i\}$ to be the background vector multiplets for $(U(1)_{\text{Top.}})^r$ topological symmetry, then the Lagrangian density of the $(\mathcal{T}_\Delta)^r/U(1)_K^r$ theory is

$$\mathcal{L}_{(\mathcal{T}_\Delta)^r/U(1)_K^r}(\mathcal{V}_1, \ldots, \mathcal{V}_r) = \sum_{i=1}^r \mathcal{L}_{\mathcal{T}_\Delta}(v_i) + \int d^4\theta \left( \frac{1}{4\pi} \sum_{i,j=1}^r K_{ij} \Sigma_{v_i} v_j + \frac{1}{2\pi} \sum_{i=1}^r \Sigma_{v_i} \mathcal{V}_i \right) . \tag{3.3}$$

We also consider a superpotential deformation by a collection of gauge-invariant 1/2 BPS chiral primary operators (CPOs) $\mathcal{O}_I$. Since the gauge charges $Q_{ij}$ of the chiral fields are chosen to be $\delta_{ij}$, there are no gauge-invariant CPOs made solely of chiral fields. However, there do exist disorder-type gauge-invariant CPOs constructed by dressing bare monopole operators with chiral fields.

**1/2 BPS monopole operators** A CPO in this description can be written as[6]

$$\mathcal{O}_{(\mathbf{n},\mathbf{m})} := \left( \prod_{i=1}^r \phi_i^{n_i} \right) V_\mathbf{m} , \tag{3.4}$$

---

[6]Throughout this paper we will not be careful to distinguish between a chiral primary multiplet and its scalar component.

where $V_{\mathbf{m}}$ denotes the 1/2 BPS bare monopole operator with monopole flux $\mathbf{m}$ and $\phi_i$ is the scalar field in $\Phi_i$. The $\mathbf{n} \in (\mathbb{Z}_{\geq 0})^r$ and $\mathbf{m} \in \mathbb{Z}^r$ are subject to the following conditions:

$$Q_i = n_i + \sum_j K_{ij} m_j - \frac{1}{2}(|m_i| + m_i) = 0, \quad i = 1, \ldots, r\,,$$

$$n_i m_i = 0, \quad i = 1, \ldots, r\,. \tag{3.5}$$

The $Q_i$ is the charge of $\mathcal{O}_{(\mathbf{n},\mathbf{m})}$ under the $i$-th $U(1)$ gauge symmetry so the first condition imposes gauge invariance. The second condition asserts that $\mathcal{O}_{(\mathbf{n},\mathbf{m})}$ is 1/2 BPS: the monopole operator should be purely electric ($m = 0$) or purely magnetic ($n = 0$) for each $U(1)$ factor of the gauge group.

**R-symmetry**  Denote $T_1, \ldots, T_r$ as the charges of the $U(1)_{\text{top}}^r$ topological symmetry which exists in the absence of the superpotential deformation. The theory also has a $U(1)_R$ symmetry which can be mixed with $U(1)_{\text{top}}^r$. Let $R_{\vec{\mu}}$ with $\vec{\mu} \in \mathbb{R}^r$ be the R-charge at general mixing defined as

$$R_{\vec{\mu}} = R_* + \vec{\mu} \cdot \vec{T}\,. \tag{3.6}$$

The reference R-charge of the monopole operator $R_*(V_{\mathbf{m}})$ is

$$R_*(V_{\mathbf{m}}) = \sum_i \left( k_{FR_*} m_i + \frac{1 - R_*(\Phi_i)}{2} |m_i| \right) = \frac{1}{2} \sum_i (m_i + |m_i|)\,. \tag{3.7}$$

After the superpotential deformations by $\{\mathcal{O}_I := \mathcal{O}_{(\mathbf{n}_I, \mathbf{m}_I)}\}_{I=1}^{N_{\mathcal{O}}}$, the mixing parameters are restricted to the space

$$\mathfrak{M}[K, \{\mathcal{O}_I\}] = \{\vec{\mu} \in \mathbb{R}^r \; : \; R_{\vec{\mu}}(\mathcal{O}_I) = R_*(V_{\mathbf{m}_I}) + \vec{\mu} \cdot \mathbf{m}_I = 2, \quad \forall I = 1, 2, \ldots, N_{\mathcal{O}}\}\,. \tag{3.8}$$

Assuming that the set $\{(\mathbf{n}_I, \mathbf{m}_I) \in \mathbb{Z}^{2r}\}_{I=1}^{N_{\mathcal{O}}}$ is linearly independent, $\mathfrak{M}$ is an $(r - N_{\mathcal{O}})$-dimensional affine subspace of $\mathbb{R}^r$. The theory $\mathcal{T}[K, \{\mathcal{O}_I\}]$ has $U(1)^{r-N_{\mathcal{O}}}$ flavor symmetries commuting with the $\mathcal{N} = 2$ supersymmetry and the space $\mathfrak{M}$ parametrizes mixing between these and the R-symmetry.

In order to identify a candidate theory $\mathcal{T}[K, \{\mathcal{O}_I\}]$ which flows to an IR $\mathcal{N} = 4$ rank-0 SCFT, we will require that there are $(r - 1)$ 1/2-BPS CPOs in the superpotential $\mathcal{W}$, i.e. $N_{\mathcal{O}} = r - 1$. This is because an $\mathcal{N} = 4$ rank-0 theory should possess only one (non-R) $U(1)$ flavor symmetry when described in terms of an $\mathcal{N} = 2$ subalgebra.[7] In this case, the affine subspace is one-dimensional and can be represented as

$$\mathfrak{M}[K, \{\mathcal{O}_I\}_{I=1}^{r-1}] = \{\vec{\mu} = \vec{\mu}_0 + \nu \vec{a} \; : \; \vec{\nu} \in \mathbb{R}\}\,. \tag{3.9}$$

The $\vec{\mu}_0$ is chosen such that $R_{\vec{\mu}_0}$ is the superconformal R-charge of the theory $\mathcal{T}[K, \{\mathcal{O}_I\}_{I=1}^{r-1}]$ which is determined by F-maximization as explained below. The vector $\vec{a}$ is orthogonal to the monopole fluxes $\mathbf{m}_I$ of $\mathcal{O}_I$ and its overall normalization is fixed by requiring that it is a primitive element in $\mathbb{Z}^r$ with first non-zero component positive. The theory $\mathcal{T}[K, \{\mathcal{O}_I\}_{I=1}^{r-1}]$ has $U(1)_A$ flavor symmetry with charge $A$ given by

$$A = \vec{a} \cdot \vec{T} = \sum_{i=1}^r a_i T_i\,. \tag{3.10}$$

If SUSY enhancement occurs, the $U(1)_A \times U(1)_R$ is expected to enhance to $SO(4)_R = SU(2)^H \times SU(2)^C$ R-symmetry. We identify

$$A = J_3^C - J_3^H\,, \tag{3.11}$$

---

[7]However, it is possible for some $\mathcal{T}[K, \{\mathcal{O}_I\}_{I=1}^{N_{\mathcal{O}}}]$ with $N_{\mathcal{O}} < r - 1$ flow to an $\mathcal{N} = 4$ rank-0 SCFT, with some UV symmetries decoupling in the IR. We will discuss such exceptional cases in section 4.3.

and define the embedding as

$$R_\nu := R_{\vec{\mu}_0} + \nu A = (J_3^C + J_3^H) + \nu(J_3^C - J_3^H) \,. \tag{3.12}$$

Here $J_3^H/J_3^C$ are the Cartan generators of $SU(2)^H/SU(2)^C$ normalized as $J_3 \in \frac{1}{2}\mathbb{Z}$.

## 3.2 Conditions on supersymmetric partition functions

The supersymmetric partition functions and indices provide strong probes of a theory's IR dynamics. Here we collect a list of supersymmetric observables which we compute in order to identify candidate theories $T[K, \{\mathcal{O}_I\}]$ that flow to rank-zero theories.

**Superconformal index** The main supersymmetric observable we compute is the superconformal index. For the theory $\mathcal{T}[K, \{\mathcal{O}_I\}_{I=1}^{r-1}]$, the index is defined by

$$\mathcal{I}_{\text{sci}}(q, \eta, \nu) = \text{Tr}_{\mathcal{H}_{\text{rad}}(S^2)}(-1)^{R_\nu} q^{\frac{R_\nu}{2}+j_3}\eta^A. \tag{3.13}$$

Here, $\mathcal{H}_{\text{rad}}(S^2)$ denotes the radially quantized Hilbert-space and the index counts local operators (with signs). It can be computed by the integral formula [41, 42]

$$\mathcal{I}_{\text{sci}}(q, \eta, \nu) = \mathcal{I}_{\text{sci}}^K(q, \vec{\zeta}, \vec{\mu})|_{\vec{\mu}=\vec{\mu}_0+\nu\vec{a}, \, \zeta_i=\eta^{a_i}}, \text{ where}$$

$$\mathcal{I}_{\text{sci}}^K(q, \vec{\zeta}, \vec{\mu}) = \sum_{m_i\in\mathbb{Z}} \oint \prod_{i=1}^r \frac{dz_i}{2\pi i z_i} \prod_{i,j=1}^r z_i^{K_{ij}m_j} \prod_{i=1}^r \left(\mathcal{I}_\Delta(m_i, z_i)(\zeta_i(-q^{1/2})^{\mu_i})^{m_i}\right) \,. \tag{3.14}$$

Here $\mathcal{I}_\Delta(m, z)$ is the tetrahedron index [43], which computes the generalized index of $\mathcal{T}_\Delta$.

**F-maximization** The IR superconformal R-charge $\vec{\mu}_0$ of $\mathcal{T}[K, \{\mathcal{O}_I\}]$ is determined by the fact that

$$F(\vec{\mu}) := -\log\left|\mathcal{Z}_{S_{b=1}^3}^K(\vec{\mu})\right| \tag{3.15}$$

is maximized at $\vec{\mu} = \vec{\mu}_0$ in the space of possible mixings $\mathfrak{M}[K, \{\mathcal{O}_I\}]$ [44–46]. $\mathcal{Z}_{S_b^3}^K$ is the squashed three-sphere partition function (with squashing parameter $b$) which can be written as the integral [47]:

$$\mathcal{Z}_{S_b^3}^K(\vec{\mu}) = \int \prod_{i=1}^r \frac{dZ_i}{\sqrt{2\pi\hbar}} \exp\left(\frac{\vec{Z}^T K\vec{Z} + 2\vec{Z}\cdot\vec{W}}{2\hbar}\right) \prod_{i=1}^r \psi_\hbar(Z_i)\Bigg|_{\vec{W}=(i\pi+\frac{\hbar}{2})\vec{\mu}} \,. \tag{3.16}$$

Here $\hbar = 2\pi i b^2$, and $\psi_\hbar(Z)$ denotes the quantum dilogarithm function [40, 48]. An alternative and more efficient method for computing $F(\vec{\mu})$ using Bethe vacua is summarized in Appendix A.

Given a theory $\mathcal{T}[K, \{\mathcal{O}_I\}]$, we can compute each of the observables listed above and determine if the theory meets the following criteria:

(a) There are $(r-1)$ linearly independent[8] 1/2 BPS CPOs $\{\mathcal{O}_I = \mathcal{O}_{(\mathbf{n}_I, \mathbf{m}_I)}\}_{I=1}^{r-1}$ satisfying (3.5).

   Note that it is possible for $\mathcal{T}[K, \{\mathcal{O}_I\}_{I=1}^{N_\mathcal{O}}]$ with $N_\mathcal{O} < r-1$ to flow to an $\mathcal{N}=4$ rank-0 SCFT due to excess $U(1)$ flavor symmetries decoupling in the IR. For simplicity, we focus on the case with $N_\mathcal{O} = r-1$.

(b) The superconformal R-charge $R_{\vec{\mu}_0}$ should satisfy the condition $\vec{\mu}_0 \in (\frac{1}{2}\mathbb{Z})^r$.

   If SUSY enhancement occurs, the $\mathcal{N}=2$ superconformal R-charge $R_{\nu=0} = R_{\vec{\mu}_0} = R_* + \vec{\mu}_0 \cdot \vec{T}$ can only take values in $\frac{1}{2}\mathbb{Z}$ since $R_{\nu=0} = J_3^C + J_3^H$. Taking into account the fact that $R_*$ and $T_i$ are integer-valued, we arrive at the condition above. This condition is restrictive as $\vec{\mu}_0$ is determined by extremizing $|\mathcal{Z}_{S_{b=1}^3}^K(\vec{\mu})|$, which is in general a transcendental function of $\vec{\mu}$.

---

[8]The set $\{(\mathbf{n}_I, \mathbf{m}_I) \in \mathbb{Z}^{2r}\}_{I=1}^{r-1}$ is linearly independent.

(c) The superconformal index satisfies

$$
\begin{aligned}
(i) \; & \mathcal{I}_{\mathrm{sci}}(q, \nu = \pm 1, \eta = 1) = 1 \; , \\
(ii) \; & \mathcal{I}_{\mathrm{sci}}(q, \nu = 0, \eta) \neq 1 \; .
\end{aligned}
\tag{3.17}
$$

If $\mathcal{T}[K, \{\mathcal{O}_I\}_{I=1}^{r-1}]$ flows to an $\mathcal{N} = 4$ SCFT, $\mathcal{I}_{\mathrm{sci}}(q, \nu = \pm 1, \eta = 1)$ only receives contributions from Coulomb/Higgs branch operators and their descendants [49]. Thus the index must be trivial in this limit if the IR SCFT is rank-0. Condition $(ii)$ ensures that the IR theory is a non-trivial SCFT with no mass gap. We examine the gapped cases in Section 4.2.

When satisfied these conditions provide strong evidence that the theory flows to an $\mathcal{N} = 4$ rank-zero SCFT.

In Section 4, we list the theories $\mathcal{T}[K, \{\mathcal{O}_I\}]$ which have a superconformal index satisfying conditions (a), (b), and (c). It is also possible to compute other supersymmetric observables, such as twisted supersymmetric partition functions on Seifert manifolds, to provide further consistency conditions. As discussed in detail in Appendix A.3, one can extract the modular data of the boundary VOA from the partition functions and this must satisfy a number of non-trivial relations. In particular, the fibering operator $\widetilde{\mathcal{F}}(u, v)$ evaluated at the Bethe vacua $u = \hat{u}$ are identified with the diagonal components of the modular $T$ matrices (see (A.21) and (A.22)), which leads to the condition

$$
\widetilde{\mathcal{F}}(\hat{u}, v)^N = 1 \; ,
\tag{3.18}
$$

for some integer $N$ and for all vacua $u = \hat{u}$. This gives a direct connection to Nahm's conjecture as summarized in Section 2.

## 3.3 Simple lines

Suppose that an $\mathcal{N} = 2$ gauge theory $\mathcal{T}[K, \{\mathcal{O}_I\}]$ flows to an $\mathcal{N} = 4$ rank-zero SCFT. Then there exists a distinguished set of line operators $\{L_1^{\mathrm{UV}}, \cdots L_d^{\mathrm{UV}}\}$ in the UV gauge theory which flow to simple lines (i.e. anyons) in the IR topologically twisted field theory. In this section, we focus on the case where UV line operators take the form of supersymmetric Wilson loops $W_{\vec{Q}}$ with charge $\vec{Q} = (Q_1, \ldots, Q_r)$, and denote by $W_{A/B}^{\mathrm{sim}}$ the collection of loops which map to simple lines. The two sets $W_A^{\mathrm{sim}}$ and $W_B^{\mathrm{sim}}$ always contain the trivial Wilson line, $1 = W_{\vec{Q}=\vec{0}}$, which corresponds to the trivial simple line in the twisted TFT. Below we will list several necessary conditions for a UV operator $W_{\vec{Q}}$ to become a simple object in the IR.

The conditions we require can be written in terms of the superconformal index with insertion of SUSY Wilson loop operators. This can be computed by multiplying the following factor in the integrand of (3.14) [43, 50]

$$
W_{\vec{Q}}^{\pm} = \prod_{i=1}^{r} (q^{m_i/2} z_i^{\pm 1})^{Q_i} \; .
\tag{3.19}
$$

In the supersymmetric $S^2 \times S^1$ background which computes the superconformal index, there are two distinguished points where one can insert a supersymmtric Wilson loop. One at the north pole of $S^2$ wrapping the $S^1$ fiber, and another at the south pole wrapping the fiber in the opposite direction. We call these $W^+$ and $W^-$ respectively and denote the superconformal index with insertions of such Wilson loops by ($\epsilon_i \in \{\pm\}$)

$$
\langle W_{\vec{Q}_1}^{\epsilon_1} W_{\vec{Q}_2}^{\epsilon_2} \ldots W_{\vec{Q}_n}^{\epsilon_n} \rangle_{\mathrm{sci}}(q, \eta, \nu) \; .
\tag{3.20}
$$

In order to test whether a UV loop $W_{\vec{Q}}$ flows to a simple line in the IR A-twisted TFT we scan over integer charge vectors $\vec{Q}$ and search for $W_{\vec{Q}}$ which satisfy[9]

$$
\begin{aligned}
&(i) \ \langle W_{\vec{Q}}^{\pm} \rangle_{\text{sci}}(q, \eta = 1, \nu = -1) = 0 \ , \ \text{ or } \ \pm q^{\mathbb{Z}/2} \ , \\
&(ii) \ \langle W_{\vec{Q}}^{+} W_{\vec{Q}}^{-} \rangle_{\text{sci}}(q, \eta = 1, \nu = -1) = 1 \ .
\end{aligned} \tag{3.21}
$$

The first condition follows from the fact that the $S^2 \times S^1$ partition function for a semi-simple TFT with a simple line insertion either vanishes or is equal to 1 if we insert the identity line. The second condition follows from the fact that the fusion between a simple object $\mathbf{L}$ and its orientation reversal $\overline{\mathbf{L}}$ always contains a single 1 (trivial object), i.e.,

$$
\mathbf{L} \times \overline{\mathbf{L}} \sim 1 + (\text{non-trivial simple objects}) \ . \tag{3.22}
$$

More concretely, we have

$$
\begin{aligned}
&\mathcal{Z}[S^2 \times S^1 \text{ with two simple objects } \mathbf{L}_\alpha \text{ and } \mathbf{L}_\beta \text{ along the } S^1] \\
&= \sum_\gamma S_{0\gamma}^2 \frac{S_{\alpha\gamma}}{S_{0\gamma}} \frac{S_{\beta\gamma}}{S_{0\gamma}} = (S^2)_{\alpha\beta} \\
&= \begin{cases} 1, & \text{if } \mathbf{L}_\alpha = \overline{\mathbf{L}}_\beta \\ 0, & \text{otherwise} \end{cases}
\end{aligned} \tag{3.23}
$$

Here $S_{\alpha\beta}$ is the modular S-matrix and $(S^2)_{\alpha\beta} = C_{\alpha\beta}$ is the charge-conjugation matrix. If a UV Wilson loop maps to a sum of multiple non-trivial simple objects in the IR, it satisfies condition $(i)$ but not condition $(ii)$. It is also possible that two different UV Wilson loops, $W_{\vec{Q}_1}$ and $W_{\vec{Q}_2}$, map to the same simple object in the IR. In that case, we expect

$$
\langle W_{\vec{Q}_1}^{+} W_{\vec{Q}_2}^{-} \rangle_{\text{sci}}(q, \eta = 1, \nu = -1) = \pm q^{\mathbb{Z}/2} \ . \tag{3.24}
$$

In general, a UV line operator $L_i^{UV}$ that maps to an IR simple line may not correspond to any supersymmetric Wilson loop. Thus we expect that this procedure gives only a subset of the simple objects in the IR twisted theory.

## 3.4 Half-index and modular functions

Finally, we consider the half-index of the gauge theory decorated by the supersymmetric line operator $L_i^{UV}$, which is defined as [51–54]

$$
I_{\text{half}}^{L_i}(q, \nu, \eta) := \text{Tr}_{\mathcal{H}^{L_i}(D_2; \mathbb{B})} (-1)^{R_\nu} q^{R_\nu/2 + j_3} \eta^A. \tag{3.25}
$$

The trace is taken over the Hilbert space on the boundary torus with a specific boundary condition $\mathbb{B}$ and in the presence of the line operator $L_i^{UV}$.

In order to make contact with the Nahm sum formula, we impose Dirichlet boundary conditions ($\mathcal{D}$) for all $\mathcal{N} = 2$ vector multiplets and deformed Dirichlet boundary conditions ($D_c$) for all $\mathcal{N} = 2$ chiral multiplets. The half-index of $\mathcal{T}[K, \{\mathcal{O}_I\}]$ with the insertion of a supersymmetric Wilson loop $W_{\vec{Q}}$ then reads [54]

$$
I_{\text{half}}^{W_{\vec{Q}}}(q, \nu, \eta) = \sum_{m \in \mathbb{N}^r} \left. \frac{q^{\frac{1}{2} m^t K m} \eta^{-a^t m} (-q^{1/2})^{-\mu^t m} q^{-Q^t m}}{(q)_{m_1} \dots (q)_{m_r}} \right|_{\mu \to \mu_0 + \nu a} , \tag{3.26}
$$

---

[9]More precisely, the conditions for $W_{\vec{Q}}$ to become a simple object, up to a factor of $U(1)_R$ flavor Wilson loop which contributes $(-q^{\pm 1/2})^{Q_R \in \mathbb{Z}}$ to $\langle W_{\vec{Q}}^{\pm} \rangle_{\text{sci}}$. These contributions cancel in cancel in $\langle W_{\vec{Q}}^{+} W_{\vec{Q}}^{-} \rangle_{\text{sci}}$.

with the Pochhammer symbol defined as $(q)_n := \prod_{i=1}^{n}(1-q^i)$. If this boundary condition is compatible with the topological A-twist in the infrared [10], we can identify this specialization of the half-index with the character of the boundary VOA of the A-twisted theory:

$$q^{\Delta} I_A[W_{\vec{Q}}] := q^{\Delta} I_{\text{half}}^{W_{\vec{Q}}}(q, \nu = -1, \eta = 1) = \sum_{m \in \mathbb{N}^r} \frac{q^{\frac{1}{2} m^t K m + \Delta}(-q^{1/2})^{-(\mu_0 - a)^t m} q^{-Q^t m}}{(q)_{m_1} \ldots (q)_{m_r}} . \qquad (3.27)$$

If $\mathcal{T}[K, \{\mathcal{O}_I\}]$ flows to a rank-zero SCFT and $W_{\vec{Q}}$ maps to a simple object in the IR $A$-twisted theory, it is reasonable to conjecture that $(3.27)$ is a modular function with a proper choice of $\Delta \in \mathbb{Q}$. We can make a similar conjecture for the $B$-twist, and two sets $W_A^{\text{sim}}$ and $W_B^{\text{sim}}$ are related to each other by

$$W_B^{\text{sim}} = \{W_{(\vec{a} - \vec{Q})} \; : \; W_{\vec{Q}} \in W_A^{\text{sim}}\} . \qquad (3.28)$$

Namely, the two sets of modular functions in A/B-twisting are identical.

For a half-index to have nice properties under modular transformations, it is natural to impose (NS, NS) boundary conditions on the boundary torus. The definition of the half-index $(3.25)$ graded by $(-1)^R$ aligns with this choice. Further, the factor $(-1)^{(\mu_0 - a)^t m}$ in the half-index, which does not appear in the original Nahm sum formula $(1.3)$, also arises from this choice of spin structure. This implies we are working with a generalization of Nahm's conjecture, as stated at the end of Section 2 and discussed in Appendix $A.4$. In Section 4, we will provide explicit examples of RCFT characters that can be expressed in this modified Nahm sum form.

## 4 Summary of results

### 4.1 A family of rank-zero theories

In this section, we summarize the classes of candidate rank-zero theories whose partition functions meet the criteria stated in Section 3. For each candidate, we also identify the set $W_A^{\text{sim}}$ of UV Wilson loops which flow to simple objects in the IR A-twisted theory based on the criteria in Section 3.3. Through these examples, we confirm the conjecture in the previous section by expressing $I_A[W_{\vec{Q}}]$ in terms of the known characters of RCFTs. For the naming convention and explicit expressions of the characters, refer to Appendix B. The identification of the indices with known RCFT characters are conjectures based on the $q$-series expansion except for a few cases with known Nahm sum representations.

The search scanned all positive definite integer $K$-matrices for $r = 1, 2, 3$ with entries between $-17$ and $17$. To avoid redundancy, we exclude candidates that are related to those with lower $r$ through basic mirror duality, or those which are constructed via direct products of other candidates. The basic mirror duality is expressed as[40, 55]

$$\frac{\mathcal{T}_{\Delta}}{U(1)_1} \simeq \mathcal{T}_{\Delta} . \qquad (4.1)$$

Under this duality, a gauge-invariant monopole operator $V_{n \geq 0}$ is mapped to $\phi^n$. By deforming the two theories with a superpotential $V_2 \simeq \phi^2$ (i.e. complex mass deformation) we find

$$\left(\frac{\mathcal{T}_{\Delta}}{U(1)_1} \text{ with a superpotential } \mathcal{W} = V_2\right) \simeq (\text{an almost trivially gapped theory}) . \qquad (4.2)$$

---

[10]This is in general a very non-trivial condition due to the existence of superpotentials

This implies that

$$\mathcal{T}\left[K_{r+1} = \begin{pmatrix} K_r & \mathbf{0}_r \\ \mathbf{0}_r & 1 \end{pmatrix}, \{V_{(\mathbf{0}_r,2)}, \{\tilde{\mathcal{O}}_I\}_{I=1}^{r-1}\}\right] \simeq \mathcal{T}[K_r, \{\mathcal{O}_I\}_{I=1}^{r-1}] . \tag{4.3}$$

Here, $\tilde{\mathcal{O}}_I$ shares the same $(n_i, m_i)$ as $\mathcal{O}_I$ for $1 \leq i \leq r-1$, but has $(n_r, m_r) = (0,0)$. At the level of the half-index, the duality can be understood from the following equality:

$$\sum_{m \in \mathbb{Z}_{\geq 0}} \frac{q^{\frac{m^2}{2}}}{(q)_m} = \prod_{n=0}^{\infty} (1 + q^{n+1/2}) = q^{1/48} \chi_F(q) . \tag{4.4}$$

Here $\chi_F$ is the character of the free Majorana fermion theory defined as

$$\begin{aligned}
\chi_F(q) &= q^{-1/48} \prod_{n=0}^{\infty} (1 + q^{n+1/2}) \\
&= q^{-1/48}(1 + q^{1/2} + q^{3/2} + q^2 + q^{5/2} + q^3 + q^{7/2} + 2q^4 + 2q^{9/2} + 2q^5 + \dots) .
\end{aligned} \tag{4.5}$$

In the bulk-boundary correspondence, the 2D RCFT corresponds to the 3D minimal invertible spin TQFT $SO(1)_1$ [11], which is an almost trivial theory.

### 4.1.1 $r = 1$

We find one theory which meets all the conditions. This theory flows to the minimal rank-zero SCFT, which was first found in [57]. We call it $\mathcal{T}_{\min}$.

1-1 $K = (2) = C(A_1)$,

$$\begin{aligned}
&\{\mathcal{O}_I\} = \emptyset, \quad \vec{\mu}_0 = (-1), \quad \vec{a} = (1), \quad W_{\text{sim}} = \{1, W_1\}, \\
&I_A[1] = q^{-11/60} \chi_{(1,1)}^{M(2,5)}, \quad I_A[W_1] = q^{1/60} \chi_{(1,2)}^{M(2,5)}, \\
&\mathcal{I}_{\text{sci}} = 1 - q - \left(\eta + \frac{1}{\eta}\right) q^{3/2} - 2q^2 - \left(\eta + \frac{1}{\eta}\right) q^{5/2} - 2q^3 - \left(\eta + \frac{1}{\eta}\right) q^{7/2} + \cdots .
\end{aligned}$$

### 4.1.2 $r = 2$

In addition to the examples which are direct products of two $r = 1$ theories, we have three candidate rank-zero theories.

2-1 $K = \begin{pmatrix} 2 & -1 \\ -1 & 1 \end{pmatrix} = C(T_2)$,

$$\begin{aligned}
&\{\mathcal{O}_I\} = \{(\phi_1)^2 V_{(0,2)}\}, \quad \vec{\mu}_0 = (-1,0), \quad \vec{a} = (1,0), \quad W_A^{\text{sim}} = \{1, W_{(1,0)}\}, \\
&I_A[1] = q^{-19/96} \chi_F \chi_{(1,1)}^{SM(2,8)}, \quad I_A[W_{(1,0)}] = q^{5/96} \chi_F \chi_{(1,3)}^{SM(2,8)}, \\
&\mathcal{I}_{\text{sci}} = 1 - q^{1/2} - \left(1 + \eta + \frac{1}{\eta}\right) q - \left(2 + \eta + \frac{1}{\eta}\right) q^{3/2} - \left(2 + \eta + \frac{1}{\eta}\right) q^2 - q^{5/2} + \cdots .
\end{aligned}$$

This example is studied in [32]. In particular, it was argued that the supersymmetry at the fixed point enhances to $\mathcal{N} = 5$. Based on various partition function calculations, it is conjectured that this is dual to $\mathcal{N} = 3$ $SU(2)_{k=2}$ Chern-Simons theory coupled to a half hypermultiplet and a half twisted hypermultiplet both in the fundamental representation.

---

[11]We used the notation in e.g., [56]

2-2 $K = \begin{pmatrix} 2 & 1 \\ 1 & 1 \end{pmatrix} = C(T_2)^{-1}$ ,

$$\{\mathcal{O}_I\} = \{V_{(2,-2)}\}, \quad \vec{\mu}_0 = (-1,-1), \quad \vec{a} = (1,1), \quad W_A^{\text{sim}} = \{1, W_{(1,1)}\},$$

$$I_A[1] = q^{-7/32}\chi_{(1,1)}^{SM(2,8)}, \quad I_A[W_{(1,1)}] = q^{1/32}\chi_{(1,3)}^{SM(2,8)},$$

$$\mathcal{I}_{\text{sci}} = 1 - q^{1/2} - \left(1 + \eta + \frac{1}{\eta}\right)q - \left(2 + \eta + \frac{1}{\eta}\right)q^{3/2} - \left(2 + \eta + \frac{1}{\eta}\right)q^2 - q^{5/2} + \cdots .$$

This theory is dual to 2-1, which can be inferred from the relation $K \leftrightarrow K^{-1}$.

2-3 $K = \begin{pmatrix} 4 & 2 \\ 2 & 2 \end{pmatrix} = C(A_1) \otimes C(T_2)^{-1}$ ,

$$\{\mathcal{O}_I\} = \{V_{(-1,2)}\}, \quad \vec{\mu}_0 = (-2,-1), \quad \vec{a} = (2,1), \quad W_A^{\text{sim}} = \{1, W_{(1,1)}, W_{(2,1)}\},$$

$$I_A[1] = q^{-17/42}\chi_{(1,1)}^{M(2,7)}, \quad I_A[W_{(1,1)}] = q^{-5/42}\chi_{(1,2)}^{M(2,7)}, \quad I_A[W_{(2,1)}] = q^{1/42}\chi_{(1,3)}^{M(2,7)},$$

$$\mathcal{I}_{\text{sci}} = 1 - q - \left(\eta + \frac{1}{\eta}\right)q^{3/2} - 2q^2 - \eta q^{5/2} + \left(\frac{1}{\eta^2} - 1\right)q^3 + \left(\frac{1}{\eta} - \eta\right)q^{7/2} + \frac{1}{\eta^2}q^4 + \cdots .$$

This example is discussed in [19, 21, 22] in detail, where it is argued that the boundary condition indeed supports the $M(2,7)$ Virasoro minimal model.

### 4.1.3   $r = 3$

In addition to the examples which are direct products of $r = 1$ and $r = 2$ theories, we find 23 candidate rank-zero theories. Many of these theories have identical superconformal indices and partition functions, which is a strong signal that they flow to the same fixed point. Below we organize the theories by duality class.

**Class 1: $\mathcal{T}_1(= \mathcal{T}_{\mathbf{min}})$** The following list of theories have the superconformal index

$$\mathcal{I}_{\text{sci}} = 1 - q - \left(\eta + \frac{1}{\eta}\right)q^{3/2} - 2q^2 - \left(\eta + \frac{1}{\eta}\right)q^{5/2} - 2q^3 - \left(\eta + \frac{1}{\eta}\right)q^{7/2} + \cdots , \qquad (4.6)$$

which is the same as that of $\mathcal{T}_{\text{min}}$. There exists a gauge theory description whose (deformed) Dirichlet half-index reproduces the characters of $M(2,5)$ or the simple affine VOA $L_1(\mathfrak{osp}(1|2))$ (with a specialization of the Jacobi variable) up to a free fermion factor $\chi_F$.

3-1 $K = \begin{pmatrix} 2 & -1 & -1 \\ -1 & 2 & 0 \\ -1 & 0 & 1 \end{pmatrix} = C(T_3)$ ,

$$\{\mathcal{O}_I\} = \{(\phi_1)^2 V_{(0,0,2)}, (\phi_3)V_{(1,1,0)}\}, \quad \vec{\mu}_0 = (-1,1,0), \quad \vec{a} = (1,-1,0), \quad W_A^{\text{sim}} = \{1, W_{(1,-1,0)}\},$$

$$I_A[1] = q^{3/80}\chi_F(q)\chi^{osp(1|2)_1}[\mathbf{1}](q,x)|_{x=-q^{1/2}},$$

$$I_A[W_{(1,-1,0)}] = q^{27/80}\chi_F(q)\chi^{osp(1|2)_1}[\mathbf{M}](q,x)|_{x=-q^{1/2}}.$$

Here $\chi^{osp(1|2)_1}[\mathbf{1}]$ and $\chi^{osp(1|2)_1}[\mathbf{M}]$ are the supercharacters of the vacuum module $\mathbf{1}$ and a non-vacuum module $\mathbf{M}$ of $L_1(\mathfrak{osp}(1|2))$. (See appendix B for the notation.)

3-2 $K = \begin{pmatrix} 2 & -1 & -1 \\ -1 & 2 & 1 \\ -1 & 1 & 1 \end{pmatrix}$ ,

    1) $\{\mathcal{O}_I\} = \{V_{(0,1,-1)}, V_{(1,1,0)}\}, \ \vec{\mu}_0 = (-1,1,0), \ \vec{a} = (1,-1,-1), \ W_A^{\mathrm{sim}} = \{1, W_{(1,-1,-1)}\},$
        $I_{\mathrm{A}}[1] = q^{-11/60} \chi_{(1,1)}^{M(2,5)}, \quad I_{\mathrm{A}}[W_{(1,-1,-1)}] = q^{1/60} \chi_{(1,2)}^{M(2,5)},$

    2) $\{\mathcal{O}_I\} = \{V_{(0,1,-1)}, V_{(2,2,0)}\}, \ \vec{\mu}_0 = (-1,0,-1), \ \vec{a} = (1,-1,-1), \ W_A^{\mathrm{sim}} = \{1, W_{(1,-1,-1)}\},$
        $I_{\mathrm{A}}[1] = q^{-11/60} \chi_{(1,1)}^{M(2,5)}, \quad I_{\mathrm{A}}[W_{(1,-1,-1)}] = q^{1/60} \chi_{(1,2)}^{M(2,5)},$

    3) $\{\mathcal{O}_I\} = \{V_{(0,2,-2)}, V_{(1,1,0)}\}, \ \vec{\mu}_0 = (1,-1,-1), \ \vec{a} = (1,-1,-1), \ W_A^{\mathrm{sim}} = \{1, W_{(1,-1,-1)}\},$
        $I_{\mathrm{A}}[1] = q^{19/60} \chi^{osp(1|2)} [\mathbf{M}]] (q,x)|_{x=-q^{1/2}},$
        $I_{\mathrm{A}}[W_{(1,-1,-1)}] = q^{1/60} \chi^{osp(1|2)} [\mathbf{1}]] (q,x)|_{x=-q^{1/2}}.$

3-3 $K = \begin{pmatrix} 2 & 1 & -1 \\ 1 & 2 & 0 \\ -1 & 0 & 1 \end{pmatrix}$ ,

    $\{\mathcal{O}_I\} = \{(\phi_1)^2 V_{(0,0,2)}, V_{(-1,1,-1)}\}, \ \vec{\mu}_0 = (-1,0,0), \ \vec{a} = (1,1,0), \ W_A^{\mathrm{sim}} = \{1, W_{(1,1,0)}\},$
    $I_{\mathrm{A}}[1] = q^{-13/80} \chi_F \chi_{(1,1)}^{M(2,5)}, \quad I_{\mathrm{A}}[W_{(1,1,0)}] = q^{3/80} \chi_F \chi_{(1,2)}^{M(2,5)}.$

3-4 $K = \begin{pmatrix} 2 & 1 & 1 \\ 1 & 2 & 1 \\ 1 & 1 & 1 \end{pmatrix}$ ,

    1) $\{\mathcal{O}_I\} = \{V_{(0,1,-1)}, V_{(1,0,-1)}\}, \ \vec{\mu}_0 = (-1,-1,-2), \ \vec{a} = (1,1,1), \ W_A^{\mathrm{sim}} = \{1, W_{(1,1,1)}\},$
        $I_{\mathrm{A}}[1] = q^{-11/60} \chi_{(1,1)}^{M(2,5)}, \quad I_{\mathrm{A}}[W_{(1,1,1)}] = q^{1/60} \chi_{(1,2)}^{M(2,5)},$

    2) $\{\mathcal{O}_I\} = \{V_{(0,1,-1)}, V_{(2,0,-2)}\}, \ \vec{\mu}_0 = (-1,0,-1), \ \vec{a} = (1,1,1), \ W_A^{\mathrm{sim}} = \{1, W_{(1,1,1)}\},$
        $I_{\mathrm{A}}[1] = q^{-11/60} \chi_{(1,1)}^{M(2,5)}, \quad I_{\mathrm{A}}[W_{(1,1,1)}] = q^{1/60} \chi_{(1,2)}^{M(2,5)},$

    3) $\{\mathcal{O}_I\} = \{V_{(0,2,-2)}, V_{(1,0,-1)}\}, \ \vec{\mu}_0 = (0,-1,-1), \ \vec{a} = (1,1,1), \ W_A^{\mathrm{sim}} = \{1, W_{(1,1,1)}\},$
        $I_{\mathrm{A}}[1] = q^{-11/60} \chi_{(1,1)}^{M(2,5)}, \quad I_{\mathrm{A}}[W_{(1,1,1)}] = q^{1/60} \chi_{(1,2)}^{M(2,5)}.$

3-5 $K = \begin{pmatrix} 3 & -2 & 2 \\ -2 & 2 & -1 \\ 2 & -1 & 2 \end{pmatrix}$ ,

    $\{\mathcal{O}_I\} = \{V_{(0,2,2)}, (\phi_2) V_{(1,0,-1)}\}, \ \vec{\mu}_0 = (1,-1,0), \ \vec{a} = (1,-1,1), \ W_A^{\mathrm{sim}} = \{1, W_{(1,-1,1)}\},$
    $I_{\mathrm{A}}[1] = q^{3/80} \chi_F \chi_{(1,2)}^{M(2,5)}, \quad I_{\mathrm{A}}[W_{(1,-1,1)}] = q^{-13/80} \chi_F \chi_{(1,1)}^{M(2,5)}.$

3-6 $K = \begin{pmatrix} 3 & -1 & -1 \\ -1 & 1 & 0 \\ -1 & 0 & 1 \end{pmatrix}$,

   1) $\{\mathcal{O}_I\} = \{(\phi_1)V_{(0,0,1)}, (\phi_1)V_{(0,1,0)}\}$, $\vec{\mu}_0 = (-4,1,1)$, $\vec{a} = (1,0,0)$, $W_A^{\text{sim}} = \{1, W_{(1,0,0)}\}$,
   $I_A[1] = 0$, $\quad I_A[W_{(1,0,0)}] = 0$,

   2) $\{\mathcal{O}_I\} = \{(\phi_1)V_{(0,0,1)}, (\phi_1)^2 V_{(0,2,0)}\}$, $\vec{\mu}_0 = (-2,0,1)$, $\vec{a} = (1,0,0)$, $W_A^{\text{sim}} = \{1, W_{(1,0,0)}\}$,
   $I_A[1] = 0$, $\quad I_A[W_{(1,0,0)}] = 0$,

   3) $\{\mathcal{O}_I\} = \{(\phi_1)^2 V_{(0,0,2)}, (\phi_1)V_{(0,1,0)}\}$, $\vec{\mu}_0 = (-2,1,0)$, $\vec{a} = (1,0,0)$, $W_A^{\text{sim}} = \{1, W_{(1,0,0)}\}$,
   $I_A[1] = 0$, $\quad I_A[W_{(1,0,0)}] = 0$.

Although the computation of the superconformal indices suggests that the bulk theory is dual to $\mathcal{T}_{\text{min}}$, the deformed Dirichlet boundary condition in this description seems incompatible with the topological twists in the IR theory.

3-7 $K = \begin{pmatrix} 3 & -1 & 1 \\ -1 & 1 & 0 \\ 1 & 0 & 1 \end{pmatrix}$,

   1) $\{\mathcal{O}_I\} = \{(\phi_1)V_{(0,1,0)}, V_{(0,1,1)}\}$, $\vec{\mu}_0 = (-2,1,-1)$, $\vec{a} = (1,0,0)$, $W_A^{\text{sim}} = \{1, W_{(1,0,0)}\}$,
   $I_A[1] = 0$, $\quad I_A[W_{(1,0,0)}] = 0$,

   2) $\{\mathcal{O}_I\} = \{(\phi_1)V_{(0,1,0)}, V_{(0,2,2)}\}$, $\vec{\mu}_0 = (-4,1,-2)$, $\vec{a} = (1,0,0)$, $W_A^{\text{sim}} = \{1, W_{(1,0,0)}\}$,
   $I_A[1] = 0$, $\quad I_A[W_{(1,0,0)}] = 0$,

   3) $\{\mathcal{O}_I\} = \{(\phi_1)^2 V_{(0,2,0)}, V_{(0,1,1)}\}$, $\vec{\mu}_0 = (-1,0,0)$, $\vec{a} = (1,0,0)$, $W_A^{\text{sim}} = \{1, W_{(1,0,0)}\}$,
   $I_A[1] = q^{-17/120}\chi_F^2\, \chi_{(1,1)}^{M(2,5)}$, $\quad I_A[W_{(1,0,0)}] = q^{7/120}\chi_F^2\, \chi_{(1,2)}^{M(2,5)}$.

3-8 $K = \begin{pmatrix} 3 & 2 & -1 \\ 2 & 2 & -1 \\ -1 & -1 & 1 \end{pmatrix}$,

   $\{\mathcal{O}_I\} = \{(\phi_1)(\phi_2)V_{(0,0,1)}, V_{(2,-2,0)}\}$, $\vec{\mu}_0 = (-2,-2,1)$, $\vec{a} = (1,1,0)$, $W_A^{\text{sim}} = \{1, W_{(1,1,0)}\}$,
   $I_A[1] = 0$, $\quad I_A[W_{(1,1,0)}] = 0$.

3-9 $K = \begin{pmatrix} 3 & 2 & 1 \\ 2 & 2 & 1 \\ 1 & 1 & 1 \end{pmatrix} = C(T_3)^{-1}$,

   $\{\mathcal{O}_I\} = \{V_{(2,-2,0)}, V_{(-1,1,1)}\}$, $\vec{\mu}_0 = (-1,-1,0)$, $\vec{a} = (1,1,0)$, $W_A^{\text{sim}} = \{1, W_{(1,1,0)}\}$,
   $I_A[1] = q^{-13/80}\chi_F\, \chi_{(1,1)}^{M(2,5)}$, $\quad I_A[W_{(1,1,0)}] = q^{3/80}\chi_F\, \chi_{(1,2)}^{M(2,5)}$.

$$\underline{\text{3-10}} \ K = \begin{pmatrix} 5 & 4 & -2 \\ 4 & 4 & -2 \\ -2 & -2 & 2 \end{pmatrix},$$

    1) $\{\mathcal{O}_I\} = \{V_{(1,-1,0)}, V_{(1,0,2)}\}$, $\vec{\mu}_0 = (1,0,-1)$, $\vec{a} = (2,2,-1)$, $W_A^{\text{sim}} = \{1, W_{(2,2,-1)}\}$,

        $I_A[1] = q^{1/60} \chi_{(1,2)}^{M(2,5)}$,    $I_A[W_{(2,2,-1)}] = q^{-11/60} \chi_{(1,1)}^{M(2,5)}$,

    2) $\{\mathcal{O}_I\} = \{V_{(1,-1,0)}, V_{(2,0,4)}\}$, $\vec{\mu}_0 = (0,-1,-1)$, $\vec{a} = (2,2,-1)$, $W_A^{\text{sim}} = \{1, W_{(2,2,-1)}\}$,

        $I_A[1] = q^{1/60} \chi_{(1,2)}^{M(2,5)}$,    $I_A[W_{(2,2,-1)}] = q^{-11/60} \chi_{(1,1)}^{M(2,5)}$.

$$\underline{\text{3-11}} \ K = \begin{pmatrix} 5 & 4 & 2 \\ 4 & 4 & 2 \\ 2 & 2 & 2 \end{pmatrix},$$

    1) $\{\mathcal{O}_I\} = \{V_{(1,-1,0)}, V_{(0,-1,2)}\}$, $\vec{\mu}_0 = (-1,-2,-1)$, $\vec{a} = (2,2,1)$, $W_A^{\text{sim}} = \{1, W_{(2,2,1)}\}$,

        $I_A[1] = q^{-11/60} \chi_{(1,1)}^{M(2,5)}$,    $I_A[W_{(2,2,1)}] = q^{1/60} \chi_{(1,2)}^{M(2,5)}$,

    2) $\{\mathcal{O}_I\} = \{V_{(1,-1,0)}, V_{(0,-2,4)}\}$, $\vec{\mu}_0 = (0,-1,-1)$, $\vec{a} = (2,2,1)$, $W_A^{\text{sim}} = \{1, W_{(2,2,1)}\}$,

        $I_A[1] = q^{-11/60} \chi_{(1,1)}^{M(2,5)}$,    $I_A[W_{(2,2,1)}] = q^{1/60} \chi_{(1,2)}^{M(2,5)}$.

$$\underline{\text{3-12}} \ \text{(An infinite family)} \ K = \begin{pmatrix} a^2+1 & a^2 & a \\ a^2 & a^2 & a \\ a & a & 2 \end{pmatrix}, \quad a \in \mathbb{Z} \setminus \{0\}$$

    1) $\begin{cases} \{\mathcal{O}_I\} = \{V_{(1,-1,0)}, V_{(0,-1,a)}\}, \ \vec{\mu}_0 = (-1,-2,-1), \ \vec{a} = (a,a,1), \quad a > 0 \\ \{\mathcal{O}_I\} = \{V_{(1,-1,0)}, V_{(1,0,-a)}\}, \ \vec{\mu}_0 = (1,0,-1), \ \vec{a} = (a,a,1), \quad a < 0 \end{cases}$

        $I_A[1] = q^{-11/60} \chi_{(1,1)}^{M(2,5)}$,    $I_A[W_{(a,a,1)}] = q^{1/60} \chi_{(1,2)}^{M(2,5)}$,

    2) $\begin{cases} \{\mathcal{O}_I\} = \{V_{(1,-1,0)}, V_{(0,-2,2a)}\}, \ \vec{\mu}_0 = (0,-1,-1), \ \vec{a} = (a,a,1), \quad a > 0 \\ \{\mathcal{O}_I\} = \{V_{(1,-1,0)}, V_{(2,0,-2a)}\}, \ \vec{\mu}_0 = (0,-1,-1), \ \vec{a} = (a,a,1), \quad a < 0 \end{cases}$

        $I_A[1] = q^{-11/60} \chi_{(1,1)}^{M(2,5)}$,    $I_A[W_{(a,a,1)}] = q^{1/60} \chi_{(1,2)}^{M(2,5)}$.

**Class 2:** $(\mathcal{T}_1)^2$   This theory is expected to flow to two copies of $\mathcal{T}_{\min}$. As evidence, we can check that the superconformal index is a square of (4.6):

$$\mathcal{I}_{\text{sci}} = 1 - 2q - \left(2\eta + \frac{2}{\eta}\right) q^{3/2} - 3q^2 + \left(2 + \eta^2 + \frac{1}{\eta^2}\right) q^3 + \left(4\eta + \frac{4}{\eta}\right) q^{7/2} + \cdots . \tag{4.7}$$

$$\underline{\text{3-13}} \ K = \begin{pmatrix} 4 & 2 & 2 \\ 2 & 2 & 1 \\ 2 & 1 & 2 \end{pmatrix},$$

    $\{\mathcal{O}_I\} = \{V_{(-1,1,1)}, V_{(-1,0,2)}\}$, $\vec{\mu}_0 = (-2,-1,-1)$, $\vec{a} = (2,1,1)$, $W_A^{\text{sim}} = \{1, W_{(2,1,1)}\}$,

    $I_A[1] = q^{-11/30} \left(\chi_{(1,1)}^{M(2,5)}\right)^2$,    $I_A[W_{(2,1,1)}] = q^{1/30} \left(\chi_{(1,2)}^{M(2,5)}\right)^2$.

The result for $I_A[1]$ follows from the identity between two Nahm sums proven in section 7 of [58]:

$$\sum_{n_1,n_2,n_3 \geq 0} \frac{q^{(n_1+n_2+n_3)(n_1+n_2)+n_2(n_2+n_3)+n_3^2+n_3+n_1+2n_2}}{(q)_{n_1}(q)_{n_2}(q)_{n_3}} = \sum_{n_1,n_2 \geq 0} \frac{q^{n_1^2+n_2^2+n_1+n_2}}{(q)_{n_1}(q)_{n_2}} . \tag{4.8}$$

**Class 3:** $\mathcal{T}_1 \times U(1)_2$  The following class of theories has the superconformal index

$$\mathcal{I}_{\text{sci}} = 1 - q - \left(\eta + \frac{1}{\eta}\right)q^{3/2} - 2q^2 - \left(\eta + \frac{1}{\eta}\right)q^{5/2} - 2q^3 - \left(\eta + \frac{1}{\eta}\right)q^{7/2} + \cdots , \qquad (4.9)$$

which is the index of $\mathcal{T}_{\min}$ multiplied by that of $U(1)_2$ theory.

$\underline{\text{3-14}}$ $K = \begin{pmatrix} 4 & -2 & 2 \\ -2 & 2 & -1 \\ 2 & -1 & 2 \end{pmatrix}$, [12]

1) $\{\mathcal{O}_I\} = \{(\phi_1)V_{(0,1,1)}, V_{(-1,0,2)}\}$, $\vec{\mu}_0 = (-2, 1, -1)$, $\vec{a} = (2, -1, 1)$,
   $W_A^{\text{sim}} = \{1, W_{(1,-1,1)}, W_{(2,-1,1)}, W_{(3,-2,2)}\}$,
   $I_A[1] = q^{-47/120}\chi_1^{U(1)_2}\chi_{(1,1)}^{M(2,5)}$, $\quad I_A[W_{(1,-1,1)}] = q^{-17/120}\chi_0^{U(1)_2}\chi_{(1,1)}^{M(2,5)}$,
   $I_A[W_{(2,-1,1)}] = q^{7/120}\chi_0^{U(1)_2}\chi_{(1,2)}^{M(2,5)}$, $\quad I_A[W_{(3,-2,2)}] = q^{-23/120}\chi_1^{U(1)_2}\chi_{(1,2)}^{M(2,5)}$,

2) $\{\mathcal{O}_I\} = \{(\phi_1)V_{(0,1,1)}, V_{(-2,0,4)}\}$, $\vec{\mu}_0 = (-3, 2, -2)$, $\vec{a} = (2, -1, 1)$,
   $W_A^{\text{sim}} = \{1, W_{(1,-1,1)}, W_{(2,-1,1)}, W_{(3,-2,2)}\}$,
   $I_A[1] = -q^{-169/240}\chi_F^{-1}\chi_{(1,1)}^{M(2,5)}$, $\quad I_A[W_{(1,-1,1)}] = -q^{1/2}I_A[1]$,
   $I_A[W_{(2,-1,1)}] = I_A[W_{(3,-2,2)}] = q^{-1/240}\chi_F^{-1}\chi_{(1,2)}^{M(2,5)}$.

**Class 4:** $\mathcal{T}_2$  The following class of theories has the superconformal index

$$\mathcal{I}_{\text{sci}} = 1 - q - \left(\eta + \frac{1}{\eta}\right)q^{3/2} - 2q^2 - \eta q^{5/2} + \left(\frac{1}{\eta^2} - 1\right)q^3 + \left(\frac{1}{\eta} - \eta\right)q^{7/2} + \frac{1}{\eta^2}q^4 + \cdots , \quad (4.10)$$

which is the same as the rank-zero theory $\mathcal{T}_2$ introduced in [19, 21]. This is also the superconformal index of the theories $\mathcal{T}_{1,2}(= \mathcal{T}_2)$ and $\overline{\mathcal{T}}_{2,1}$ in [22]. The latter two theories are related by mirror symmetry

$$(\mathcal{T}_{1,2})^\vee = \overline{\mathcal{T}}_{2,1} , \qquad (4.11)$$

where $\mathcal{T}^\vee$ and $\overline{\mathcal{T}}$ denote the mirror dual and the orientation reversal of $\mathcal{T}$ respectively. It is argued in *loc. cit.* that the Dirichlet half-index of the A- and B-twist of $\mathcal{T}_{1,2}$ (with a suitable specialization) gives the characters of $L_2(\mathfrak{osp}(1|2))$ and $M(2,7)$ respectively. The Dirichlet half-index of the A- and B-twist of $\mathcal{T}_{2,1}$ reproduces the characters of $L_1(\mathfrak{osp}(1|4))$ and the minimal W-algebra $W_{1/2}^{\min}(\mathfrak{sp}(4))$, which are also realized at the left boundary of the mirror dual theory $\mathcal{T}_{1,2}$.

$\underline{\text{3-15}}$ $K = \begin{pmatrix} 2 & -1 & -1 \\ -1 & 2 & 0 \\ -1 & 0 & 2 \end{pmatrix} = C(A_3)$,

$\{\mathcal{O}_I\} = \{(\phi_2)V_{(1,0,1)}, (\phi_3)V_{(1,1,0)}\}$, $\vec{\mu}_0 = (1, -1, -1)$, $\vec{a} = (1, -1, -1)$,
$W_A^{\text{sim}} = \{1, W_{(1,-1,0)} = W_{(1,0,-1)}, W_{(1,-1,-1)}\}$,
$I_A[1] = q^{1/14}\chi^{osp(1|4)_1}[\mathbf{1}](q, x)|_{x=1}$,
$I_A[W_{(1,-1,0)}] = q^{-3/14}\chi^{osp(1|4)_1}[\mathbf{M}_1](q, x)|_{x=1}$,
$I_A[W_{(1,-1,-1)}] = q^{-5/14}\chi^{osp(1|4)_1}[\mathbf{M}_2](q, x)|_{x=1}$.

---

[12]For the second example, we find $\mathcal{Z}_{S^2 \times S^1}^{\text{top}} = 0$, although the superconformal index is equal to 1 at the A/B-twisted points. See section 5 for more discussion of this example.

3-16 $K = \begin{pmatrix} 4 & -2 & -1 \\ -2 & 2 & 1 \\ -1 & 1 & 1 \end{pmatrix}$

$\{\mathcal{O}_I\} = \{(\phi_1)V_{(0,1,-1)}, V_{(1,1,1)}\}$, $\vec{\mu}_0 = (-2,1,0)$, $\vec{a} = (2,-1,-1)$, $W_A^{\text{sim}} = \{1, W_{(1,0,0)}, W_{(2,-1,-1)}\}$,
$I_A[1] = q^{-17/42}\chi_{(1,1)}^{M(2,7)}$, $\quad I_A[W_{(1,0,0)}] = q^{-5/42}\chi_{(1,2)}^{M(2,7)}$, $\quad I_A[W_{(2,-1,-1)}] = q^{1/42}\chi_{(1,3)}^{M(2,7)}$.

3-17 $K = \begin{pmatrix} 9 & 5 & 3 \\ 5 & 4 & 1 \\ 3 & 1 & 2 \end{pmatrix}$

$\{\mathcal{O}_I\} = \{V_{(1,-1,-1)}, (\phi_2)^2 V_{(-1,0,3)}\}$, $\vec{\mu}_0 = (-2,-2,-1)$, $\vec{a} = (3,2,1)$, $W_A^{\text{sim}} = \{1, W_{(2,1,1)}, W_{(3,2,1)}\}$,
$I_A[1] = q^{-17/42}\chi_{(1,1)}^{M(2,7)}$, $\quad I_A[W_{(2,1,1)}] = q^{-5/42}\chi_{(1,2)}^{M(2,7)}$, $\quad I_A[W_{(3,2,1)}] = q^{1/42}\chi_{(1,3)}^{M(2,7)}$.

**Class 5: $\mathcal{T}_3$** The following theory has the superconformal index

$$\mathcal{I}_{\text{sci}} = 1 - q - \left(\eta + \frac{1}{\eta}\right)q^{3/2} - 2q^2 - \eta q^{5/2} + \left(\frac{1}{\eta^2} - 1\right)q^3 + \left(\frac{1}{\eta} - \eta\right)q^{7/2} + \left(\eta + \frac{2}{\eta} - \frac{1}{\eta^3}\right)q^{9/2} + \cdots .$$
$$(4.12)$$

This theory is expected to flow to the rank-zero theory called $\mathcal{T}_3$ as discussed in [19]. The topological A-twist and B-twist admits the boundary condition that supports $M(2,9)$ and $L_3(\mathfrak{osp}(1|2))$ respectively, as discussed in [21].

3-18 $K = \begin{pmatrix} 6 & 4 & 2 \\ 4 & 4 & 2 \\ 2 & 2 & 2 \end{pmatrix}$

$\{\mathcal{O}_I\} = \{V_{(0,-1,2)}, V_{(-1,2,-1)}\}$, $\vec{\mu}_0 = (-3,-2,-1)$, $\vec{a} = (3,2,1)$, $W_A^{\text{sim}} = \{1, W_{(1,1,1)}, W_{(2,2,1)}, W_{(3,2,1)}\}$,
$I_A[1] = q^{-23/36}\chi_{(1,1)}^{M(2,9)}$, $\quad I_A[W_{(1,1,1)}] = q^{-11/36}\chi_{(1,2)}^{M(2,9)}$,
$I_A[W_{(2,2,1)}] = q^{-1/12}\chi_{(1,3)}^{M(2,9)}$, $\quad I_A[W_{(3,2,1)}] = q^{1/36}\chi_{(1,4)}^{M(2,9)}$.

**Class 6:** The following theory has the superconformal index

$$\mathcal{I}_{\text{sci}} = 1 - q^{1/2} - \left(1 + \eta + \frac{1}{\eta}\right)q - \left(2 + \eta + \frac{1}{\eta}\right)q^{3/2} - \left(2 + \eta + \frac{1}{\eta}\right)q^2 - q^{5/2} + \cdots , \quad (4.13)$$

which agrees with the superconformal index of the theory $\mathcal{T}_{(2,8)}$ in [32]. The Dirichlet half-index of this theory reproduces the vacuum character of the $\mathcal{N} = 1$ super Virasoro minimal model $SM(2,8)$.

3-19 $K = \begin{pmatrix} 4 & 2 & 1 \\ 2 & 2 & 0 \\ 1 & 0 & 1 \end{pmatrix}$

$\{\mathcal{O}_I\} = \{V_{(1,-1,-1)}, (\phi_3)^2 V_{(-2,4,0)}\}$, $\vec{\mu}_0 = (-1,-1,-1)$, $\vec{a} = (2,1,1)$, $W_A^{\text{sim}} = \{1, W_{(2,1,1)}\}$,
$I_A[1] = q^{-7/32}\chi_{(1,1)}^{SM(2,8)}$, $\quad I_A[W_{(2,1,1)}] = q^{1/32}\chi_{(1,3)}^{SM(2,8)}$.

**Class 7:** The following class of theories has the superconformal index

$$\mathcal{I}_{\text{sci}} = 1 - q^{1/2} - \left(1 + \eta + \frac{1}{\eta}\right)q - \left(2 + \eta + \frac{1}{\eta}\right)q^{3/2} - q^2 + \left(2 + \eta^2 + \frac{1}{\eta^2} + 2\eta + \frac{2}{\eta}\right)q^{5/2} + \cdots , \quad (4.14)$$

which agrees with the superconformal index of the theory $\mathcal{T}_{(2,12)}$ defined in [32]. The Dirichlet half-index of the theory reproduces the vacuum character of the $\mathcal{N} = 1$ super Virasoro minimal model $SM(2,12)$.

3-20 $K = \begin{pmatrix} 2 & 1 & 1 \\ 1 & 2 & 0 \\ 1 & 0 & 2 \end{pmatrix}$

$\{\mathcal{O}_I\} = \{V_{(-2,2,2)}, V_{(2,-1,-1)}\}$, $\vec{\mu}_0 = (-1,-1,-1)$, $\vec{a} = (0,1,-1)$, $W_A^{\text{sim}} = \{1, W_{(0,1,-1)}, W_{(1,1,0)}\}$,

$I_A[1] = q^{-7/48} \chi_F^{-1} \chi_{(1,3)}^{SM(2,12)}$,     $I_A[W_{(0,1,-1)}] = q^{-7/48} \chi_F^{-1} \chi_{(1,3)}^{SM(2,12)}$,

$I_A[W_{(1,1,0)}] = q^{1/48} \chi_F^{-1} \left( -\chi_{(1,1)}^{SM(2,12)} + \chi_{(1,5)}^{SM(2,12)} \right)$.

3-21 $K = \begin{pmatrix} 4 & 2 & -1 \\ 2 & 2 & -1 \\ -1 & -1 & 1 \end{pmatrix}$,

$\{\mathcal{O}_I\} = \{(\phi_1)^2(\phi_2)^2 V_{(0,0,2)}, (\phi_3)V_{(-1,2,0)}\}$, $\vec{\mu}_0 = (-2,-1,0)$, $\vec{a} = (2,1,0)$, $W_A^{\text{sim}} = \{1, W_{(1,1,0)}, W_{(2,1,0)}\}$,

$I_A[1] = q^{-7/16} \chi_F \chi_{(1,1)}^{SM(2,12)}$,     $I_A[W_{(1,1,0)}] = q^{-5/48} \chi_F \chi_{(1,3)}^{SM(2,12)}$,     $I_A[W_{(2,1,0)}] = q^{1/16} \chi_F \chi_{(1,5)}^{SM(2,12)}$.

3-22 $K = \begin{pmatrix} 4 & 2 & 1 \\ 2 & 2 & 0 \\ 1 & 0 & 1 \end{pmatrix}$

$\{\mathcal{O}_I\} = \{V_{(2,-2,-2)}, (\phi_3)V_{(-1,2,0)}\}$, $\vec{\mu}_0 = (-2,-1,-1)$, $\vec{a} = (2,1,1)$, $W_A^{\text{sim}} = \{1, W_{(2,1,1)}\}$,

$I_A[1] = q^{-11/24} \chi_{(1,1)}^{SM(2,12)}$,     $I_A[W_{(2,1,1)}] = q^{1/24} \chi_{(1,5)}^{SM(2,12)}$.

**Class 8:**   The following theory has the superconformal index

$$\mathcal{I}_{\text{sci}} = 1 - q - \eta q^{3/2} - \left(1 - \frac{1}{\eta^2}\right) q^2 + \frac{2q^{5/2}}{\eta} + \left(3 + \eta^2 + \frac{2}{\eta^2}\right) q^3 + \left(3\eta + \frac{5}{\eta}\right) q^{7/2} + \cdots. \quad (4.15)$$

Note that this is an example where the superconformal index does not contain the term

$$-(\eta + 1/\eta)q^{3/2} \quad (4.16)$$

in the $q$-expansion. The existence of this term is a strong signal that the supersymmetry is enhanced to $\mathcal{N} = 4$, since it coincides with the contribution from the extra supercurrent multiplet. However, it is neither a necessary nor sufficient condition, as the spectrum may contain other multiplets whose contribution to the index is the same, possibly with the opposite sign.

The examples in this class provides a gauge theory description whose (deformed) Dirichlet half-index reproduces the characters of the W-algebra minimal model $W_3(3,7)$.

3-23 $K = \begin{pmatrix} 5 & -3 & 2 \\ -3 & 3 & -1 \\ 2 & -1 & 1 \end{pmatrix}$,

$\{\mathcal{O}_I\} = \{(\phi_2)V_{(1,0,-2)}, V_{(1,2,1)}\}$, $\vec{\mu}_0 = (-1,0,-1)$, $\vec{a} = (4,-3,2)$,

$W_A^{\text{sim}} = \{1, W_{(2,-1,1)}, W_{(3,-2,1)}, W_{(4,-3,2)}\}$,

$I_A[1] = I_A[W_{(4,-3,2)}] = 0$,     $I_A[W_{(2,-1,1)}] = I_A[W_{(3,-2,1)}] = 1$.

3-24 $K = \begin{pmatrix} 6 & -3 & -1 \\ -3 & 2 & 1 \\ -1 & 1 & 1 \end{pmatrix}$,

$$\{\mathcal{O}_I\} = \{(\phi_1)^2 V_{(0,1,-1)}, V_{(1,1,2)}\}, \ \vec{\mu}_0 = (-3,1,0), \ \vec{a} = (3,-1,-1),$$
$$W_A^{\text{sim}} = \{1, W_{(1,0,0)}, W_{(-1,1,0)}, W_{(3,-1,-1)}\},$$
$$I_A[1] = q^{-19/28} \chi_{(5,1,1)}^{W_3(3,7)}, \quad I_A[W_{(1,0,0)}] = q^{-1/4} \chi_{(4,2,1)}^{W_3(3,7)},$$
$$I_A[W_{(-1,1,0)}] = q^{-1} I_A[W_{(1,0,0)}], \quad I_A[W_{(3,-1,-1)}] = q^{1/28} \chi_{(3,2,2)}^{W_3(3,7)}.$$

## 4.2 Unitary TFTs

When an $\mathcal{N} = 2$ gauge theory $\mathcal{T}[K, \{\mathcal{O}_I\}]$ has a mass gap, the theory flows to a unitary TFT in the IR which supports a unitary rational VOA at the boundary. A strong signal for the theory $\mathcal{T}[K, \{\mathcal{O}_I\}]$ to have a mass gap is

$$\mathcal{I}_{\text{sci}}^K(q, \vec{\mu}, \vec{\zeta} = \vec{1}) = 1 \text{ for all } \vec{\mu} \in \mathfrak{M}[K, \{\mathcal{O}_I\}], \tag{4.17}$$

which implies that the only local operator in the unitary TFT is the identity operator. Similar to the non-unitary case, the UV gauge theory has a flavor symmetry whose rank is equal to the dimension of $\mathfrak{M}[K, \{\mathcal{O}_I\}]$. For some examples the index is 1 for all $\vec{\mu} \in \mathfrak{M}[K, \{\mathcal{O}_I\}]$, even though its dimension is non-zero. In this case, it is natural to expect that the UV flavor symmetry decouples in the IR. As a consequence, the superconformal index and the three-sphere free energy defined in (3.15) do not depend on the mixing parameter $\vec{\mu} \in \mathfrak{M}[K, \{\mathcal{O}_I\}]$.

For a UV supersymmetric Wilson loop $W_{\vec{Q}}$ to be a simple object in the IR TFT, it must satisfy conditions similar to those in (3.21):

$$\begin{aligned} (i) & \quad \langle W_{\vec{Q}}^+ \rangle_{\text{sci}}(q, \vec{\mu}, \vec{\zeta} = 1) = 0 \text{ or } \pm q^{\mathbb{Z}/2}, \\ (ii) & \quad \langle W_{\vec{Q}}^+ W_{\vec{Q}}^- \rangle_{\text{sci}}(q, \vec{\mu}, \vec{\zeta} = 1) = 1, \end{aligned} \tag{4.18}$$

for all $\vec{\mu} \in \mathfrak{M}[K, \{\mathcal{O}_I\}]$.

If $\mathcal{T}[K, \{\mathcal{O}_I\}]$, with positive definite $K$, has a mass gap and a UV loop $W_{\vec{Q}}$ flows to a simple object in the IR TFT, the half-index

$$q^\Delta I_{\text{half}}^{W_{\vec{Q}}}(q, \vec{\mu}) = \sum_{m \in \mathbb{N}^r} \frac{q^{\frac{1}{2} m^t K m + \Delta} (-q^{1/2})^{-\mu^t m} q^{-Q^t m}}{(q)_{m_1} \dots (q)_{m_r}} \tag{4.19}$$

is expected to be a modular function upon a proper choice of $\vec{\mu} \in \mathfrak{M}[K, \{\mathcal{O}_I\}]$ and $\Delta \in \mathbb{Q}$.

Although we do not perform an exhaustive search for the unitary cases, we list some examples below.

### 4.2.1 $r = 1$

There is only one example:

U1-1 $K = (1) = C(T_1), \ \{\mathcal{O}_I\} = \{V_{(2)}\}, \ \mathfrak{M}[K, \{\mathcal{O}_I\}] = \{(0)\},$

$$I_{\text{half}}(q) = \sum_{m \geq 0} \frac{q^{m^2/2}}{(q)_m} = \prod_{n=0}^{\infty} (1 + q^{n+1/2}) := q^{1/48} \chi_F. \tag{4.20}$$

The half-index coincides with the character of a free fermion given in (4.4) and (4.5).

**4.2.2** $r = 2$

U2-1 $K = \begin{pmatrix} a & 1-a \\ 1-a & a \end{pmatrix}\Big|_{a \geq 1}$, $\quad \{\mathcal{O}_I\} = \{V_{(1,1)}\}, \quad \mathfrak{M}[K, \{\mathcal{O}_I\}] = \{(0,0) + \nu(1,-1)\}_{\nu \in \mathbb{R}}$,

$$I_{\text{half}}(q, \nu; a) = \frac{q^{-\nu^2/8a}}{(q)_\infty} \sum_{m \in \mathbb{Z}} (-1)^{\nu m} q^{\frac{1}{2}a(m - \frac{\nu}{2a})^2} .$$

This relation follows from the identity

$$\sum_{m \geq 0} \frac{q^{m(m+n)}}{(q)_m (q)_{m+n}} = \frac{1}{(q)_\infty} \quad (n \in \mathbb{Z}_{\geq 0}) . \tag{4.21}$$

We find that $q^\Delta I_{\text{half}}(q, \nu; a)$ with $\Delta = \frac{\nu^2}{8a} - 1/24$ is identical to the vacuum character of $U(1)_a$ WZW model when $\nu \in 2a\mathbb{Z}$. This indicates that the IR theory is the $U(1)_a$ Chern-Simons theory.

U2-2 $K = \begin{pmatrix} a & -1 \\ -1 & 1 \end{pmatrix}\Big|_{a \geq 2}$, $\quad \{\mathcal{O}_I\} = \{\phi_1 V_{(0,1)}\}, \quad \mathfrak{M}[K, \{\mathcal{O}_I\}] = \{(0,1) + \nu(1,0)\}_{\nu \in \mathbb{R}}$,

$$I_{\text{half}}(q, \nu) = 0 .$$

This theory is dual to

    $U(1)_{a-1}$ coupled to $\Phi_1$(of charge $+1$) and $\Phi_2$ (of charge $-1$) with superpotential $\mathcal{W} = \Phi_1 \Phi_2$

    $\simeq U(1)_{a-1}$ Chern-Simons theory.

The first line follows from the basic mirror symmetry reviewed in Section 4.1. Integrating out the two chiral fields, we obtain the pure CS theory in the infrared. As discussed in [54], the deformed Dirichlet boundary condition for $\Phi_1$ breaks the supersymmetry spontaneously, unless its boundary value is at the critical point of the superpotential.

U2-3 $K = \begin{pmatrix} a+1 & a \\ a & a \end{pmatrix}\Big|_{a \geq 1}$, $\quad \{\mathcal{O}_I\} = \{V_{(1,-1)}\}, \quad \mathfrak{M}[K, \{\mathcal{O}_I\}] = \{(1,0) + \nu(1,1)\}_{\nu \in \mathbb{R}}$,

$$I_{\text{half}}(q, \nu) = 1 .$$

The half-index computation suggests that the only boundary operator that survives in this case is the identity operator. It is interesting to note that we also have [13]

$$I_{\text{half}}^{W_{\vec{Q}}}(q, \nu, \eta) = \langle W_{\vec{Q}}^+ \rangle_{\text{sci}}(q, \nu, \eta) , \quad \text{for all } \vec{Q}. \tag{4.22}$$

**4.2.3** $r = 3$

U3-1 $K = \begin{pmatrix} 2 & 1 & 1 \\ 1 & a & 2-a \\ 1 & 2-a & a \end{pmatrix}$

$\{\mathcal{O}_I\} = \{V_{(-1,1,1)}, V_{(2,-1,-1)}\}, \quad \mathfrak{M}[K, \{\mathcal{O}_I\}] = \{(0,0,0) + \nu(0,1,-1)\}_{\nu \in \mathbb{R}}$,

$$I_{\text{half}}(q, \nu; a) = \frac{q^{-\nu^2/8a}}{(q)_\infty} \sum_{m \in \mathbb{Z}} (-1)^{\nu m} q^{\frac{1}{2}a(m - \frac{\nu}{2a})^2},$$

which is same as U2-1.

---

[13]It is tempting to conjecture that the deformed Dirichlet boundary condition flows to that corresponds to "closing the puncture" of $D^2 \times S^1$ in the IR, which produces the partition function on $S^2 \times S^1$.

U3-2  $K = \begin{pmatrix} 1+a & a & -a \\ a & a & 1-a \\ -a & 1-a & a \end{pmatrix}\Big|_{a \geq 2}$

$$\{\mathcal{O}_I\} = \{V_{(0,1,1)}, \phi_3^2 V_{(2,-2,0)}\}\ ,\ \mathfrak{M}[K, \{\mathcal{O}_I\}] = \{(0,0,0) + \nu(1,1,-1)\}_{\nu \in \mathbb{R}}\ ,$$

$$\chi(q, \nu; a) = q^{\Delta} I_{\text{half}}(q, \nu; a) = \left( \frac{q^{-1/24}}{(q)_{\infty}} \sum_{m \in \mathbb{Z}} (-1)^{\nu m} q^{\frac{1}{2} a (m - \frac{\nu}{2a})^2} \right) \chi_F(q),$$

where $\Delta = q^{\nu^2/8a - 1/24 - 1/48}$. For even $\nu$, $\chi(q, \nu; a) = \chi_{\nu/2}^{U(1)_a}(q) \chi_F(q)$, which indicates that the bulk theory flows to $U(1)_a \otimes SO(1)_1$ Chern-Simons theory.

## 4.3   Comparision with Zagier's result

Let us now compare our results with those in [1]. The half-indices $I_A[W_{\vec{Q}}]$ in (3.27) and $I_{\text{half}}^{W_{\vec{Q}}}(q, \vec{\mu})$ in (4.19) correspond to the Nahm sum formula $\chi_{(A,B,C)}$ in (1.3) with the identification

$$A = K, \ B = -\frac{(\mu_0 - a - 2Q)}{2}\ (\text{or } B = -\frac{(\mu - 2Q)}{2})\ \text{and } C = \Delta\ , \tag{4.23}$$

provided that $B \in \mathbb{Z}^r$. We find that all the triplets $(A, B, C)$ in Zagier's list, summarized in Table 1 (with $B \in \mathbb{Z}^r$), appear in our classification except for two cases.

The first exception is [14]

$$K = \begin{pmatrix} 2+a & a & -a \\ a & a & 1-a \\ -a & 1-a & a \end{pmatrix} \quad (a \geq 3)\ . \tag{4.24}$$

In this case $\mathcal{T}[K, \emptyset]$ has only one primitive 1/2 BPS monopole operator $V_{(0,1,1)}$. After the superpotential deformation, the gauge theory $\mathcal{T}[K, V_{(0,1,1)}]$ flows to an $\mathcal{N} = 4$ SCFTs in the IR. The UV gauge theory has two $U(1)$ topological symmetries but the computation of superconformal index strongly suggests that one linear combintations of them acts trivially in the IR. If this happens, the example does not show up in our search due to our simplifying assumption (a) in section 3.2.

The remaining faithful $U(1)$ can be identified with the $U(1)_A$ symmetry. The superconformal R-charge and the axial $U(1)$ symmetry $A = \vec{a} \cdot \vec{T}$ is given by

$$\vec{\mu}_0 = (0,1,-1) + \alpha(1,1,-1)\ , \quad \vec{a} = (0,1,-1)\ . \tag{4.25}$$

Here $\alpha \in \mathbb{R}$ parametrize the mixing of R-symmetry with the decoupled $U(1)$ symmetry. One can check that the superconformal index is independent of $\alpha$. The half-index computation at $\alpha = 0$ becomes

$$\begin{aligned}
I_{\text{half}}(q, \nu = -1, \eta^{-1}) &= \sum_{m_1, m_2, m_3 \geq 0} \frac{q^{\frac{1}{2} m^t K m} \eta^{m_2 - m_3}}{(q)_{m_1} (q)_{m_2} (q)_{m_3}} \\
&= \left( I_{\text{half}}(q, \nu = 1, \eta)[\mathcal{T}_{\underline{1\text{-}1}}] \right) \times \left( \frac{1}{(q)_{\infty}} \sum_{m \in \mathbb{Z}} q^{am^2/2} \eta^{-m} \right)\ .
\end{aligned} \tag{4.26}$$

In the A-twist limit, the half-index becomes

$$I_{\text{half}}(q, \nu = -1, \eta = 1) = q^{\frac{7}{120}} \chi_{(1,2)}^{M(2,5)} \chi_0^{U(1)_a}\ , \tag{4.27}$$

---

[14] The $K$ with $a = 1$ and $a = 2$ corresponds to the $K$ for 3-7 and 3-14 each.

which suggests that $\mathcal{T}[K, \{V_{(0,1,1)}\}]$ is a mirror dual of $\mathcal{T}_{\underline{1\text{-}1}} \otimes U(1)_a$.

The second exception is the case

$$A = \begin{pmatrix} 2 & 1 & 1 \\ 1 & 2 & 0 \\ 1 & 0 & 2 \end{pmatrix}, \ B = \begin{pmatrix} 1 \\ 1 \\ 0 \end{pmatrix} \text{ or } \begin{pmatrix} 1 \\ 0 \\ 1 \end{pmatrix}, \ C = 5/24 \tag{4.28}$$

in Zagier's list. The Nahm sum formula with (4.28) for either choice of $B$ is

$$\chi_{(A,B,C)}(q) = \frac{1}{2}\chi_1^{U(1)_2}(q) = 1 + q + 3q^2 + 4q^3 + 7q^4 + 10q^5 + \cdots . \tag{4.29}$$

This is a modular function but we do not know if it can be identified with a character of any rational VOA.[15] It can be reproduced from the half index of $\underline{U3\text{-}1}$ with $a = 2$ at $\vec{\mu} = \vec{0}$ decorated by a supersymmetric Wilson loop of charge $\vec{Q} = -(1,1,0)$. However, when computing the superconformal index:

$$\langle W_{\vec{Q}}^+ W_{\vec{Q}}^- \rangle_{\text{sci}}(q, \eta = 1, \vec{\mu} = \vec{0}) \neq 1, \tag{4.30}$$

this implies that the UV Wilson loop $W_{\vec{Q}=-(1,1,0)}$ does not flow to a simple object in the IR.

## 5 Discussion

**Relevance of superpotentials** The gauge theory $\mathcal{T}[K, \{\mathcal{O}_I\}_{I=1}^{r-1}]$ should be understood as a sequence of renormalization group (RG) flows:

$$\mathcal{T}[K, \emptyset] \xrightarrow{\delta\mathcal{W}=\mathcal{O}_1} \mathcal{T}[K, \{\mathcal{O}_1\}] \xrightarrow{\delta\mathcal{W}=\mathcal{O}_2} \ldots \xrightarrow{\delta\mathcal{W}=\mathcal{O}_{r-1}} \mathcal{T}[K, \{\mathcal{O}_I\}_{I=1}^{r-1}] . \tag{5.1}$$

For these RG flows to make sense, the superpotential at each step must be relevant:

$$R_0^{(I-1)}(\mathcal{O}_I) < 2 \quad \text{for } I = 1, \ldots, r-1. \tag{5.2}$$

Here, $R_0^{(I-1)} = R_* + \vec{\mu}_0^{(I-1)} \cdot \vec{T}$ represents the superconformal R-charge of $\mathcal{T}[K, \{\mathcal{O}_A\}_{A=1}^{I-1}]$, which can be determined by extremizing $F(\vec{\mu})$ over $\vec{\mu} \in \mathfrak{M}[K, \{\mathcal{O}_1, \ldots, \mathcal{O}_{I-1}\}]$. Thus, in order for the fully deformed theory, $\mathcal{T}[K, \{\mathcal{O}_I\}_{I=1}^{r-1}]$, to be sensible, there should be a proper sequence of 1/2-BPS operators, $\mathcal{O}_1, \mathcal{O}_2, \ldots, \mathcal{O}_{r-1}$, which ensures that these relevance conditions are satisfied.

Verifying the relevance of the superpotential is a challenging task and will be addressed in future work. Instead, we will list inconsistencies which indirectly demonstrate that certain examples in our classification may violate this condition:

- In the cases of $\underline{3\text{-}14}$-(2) and $\underline{3\text{-}20}$, the half indices $I_A[W_{\vec{Q}}]$ contain a factor $\chi_F^{-1}$. This does not correspond to any known RCFT character in the literature.

- For the $\underline{3\text{-}14}$-(2) case, the A and B-twisted partition functions on $S^2 \times S^1$ are not equal to 1.

- In the $\underline{3\text{-}20}$ case, the half-index $I_A[W_{(1,1,0)}]$ is expressed as a sum of two characters (modulo a $\chi_F^{-1}$), even though $W_{(1,1,0)} \in W_A^{\text{sim}}$.

- Notably, both $\underline{3\text{-}14}$-(2) and $\underline{3\text{-}20}$ involve a non-primitive monopole operator: $V_{(-2,0,4)}$ in $\underline{3\text{-}14}$-(2) and $V_{(-2,2,2)}$ in $\underline{3\text{-}20}$, which we suspect to be irrelevant.

---

[15]As coefficients of $\chi_1^{U(1)_2}(q)$ are all even, the coefficients of $q$-series are all integer-valued

**UV Wilson loops which flow to sums of IR simple objects** In section 3.3, we have scanned UV Wilson loops which flow to simple objects in the IR TFT. Half-indices with Wilson loops insertions matched to RCFT characters of corresponding primaries. There are some exceptional cases where we could not find the UV Wilson loop corresponding to an IR simple object:

$$\chi_{(1,3)}^{SM(2,12)} \text{ of } \underline{3\text{-}22}, \text{ and } \chi_{(3,3,1)}^{W_3(3,7)} \text{ of } \underline{3\text{-}24}.$$

In $\underline{3\text{-}22}$, we could not find the UV Wilson loop corresponding to $\chi_{(1,3)}^{SM(2,12)}$. Instead, we find

$$I_A[W_{(1,1,0)}] = q^{-1/8}\chi_{(1,3)}^{SM(2,12)} - q^{1/24}\chi_{(1,1)}^{SM(2,12)}. \tag{5.3}$$

This relation indicates that the UV Wilson loop flows to a non-simple line in the IR, which can be written as a linear combination of two simple lines. Suppose that the deformed Dirichlet boundary condition in the UV gauge theory maps to a holomorphic boundary condition in the IR topological theory denoted by $\langle\text{hol}|$. The vacuum character of the boundary algebra can be written as

$$\chi_{\text{vac}}(q) = q^{-c/24}\langle\text{hol}|D_2 \times S^1\rangle , \tag{5.4}$$

where $\langle\text{hol}|D_2 \times S^1\rangle$ is the partition function of the IR theory on $D^2 \times S^1$ with the holomorphic boundary condition. We also denote the IR simple loops by $\{\mathbf{L}_h\}$, with $h$ representing the conformal dimension of the corresponding module. They are normalized as follows:

$$\chi_h(q) = q^{h-c/24}\langle\text{hol}|\mathbf{L}_h|D_2 \times S^1\rangle , \tag{5.5}$$

where $\langle\text{hol}|\mathbf{L}|D_2 \times S^1\rangle$ is the partition function decorated by a loop operator $\mathbf{L}$ along the $S^1$.

The result (5.3) suggests that

$$W_{(1,1,0)} \xrightarrow{\text{RG and A-twisting}} \mathbf{L}_{-\frac{1}{3}} - q^{1/2}I . \tag{5.6}$$

It is compatible with the following superconformal index computation:

$$\begin{aligned}
\langle W_{(1,1,0)}^{-}W_{(1,1,0)}^{+}\rangle_{\text{sci}} &= \langle D_2 \times S^1|(\mathbf{L}_{-\frac{1}{3}} - q^{1/2}I)^{\dagger}(\mathbf{L}_{-\frac{1}{3}} - q^{1/2}I)|D_2 \times S^1\rangle \\
&= \langle D_2 \times S^1|(\mathbf{L}_{-\frac{1}{3}}^{\dagger} - q^{-1/2}I)(\mathbf{L}_{-\frac{1}{3}} - q^{1/2}I)|D_2 \times S^1\rangle \\
&= 2 .
\end{aligned} \tag{5.7}$$

The manifold $S^2 \times S^1$ can be obtained by gluing two solid tori with opposite orientations, and its partition function can be expressed as $\langle D_2 \times S^1|D_2 \times S^1\rangle$, which is 1 for semi-simple TFTs. For two simple objects $\mathbf{L}_\alpha$ and $\mathbf{L}_\beta$, we have $\langle D_2 \times_q S^1|\mathbf{L}_\alpha^{\dagger}\mathbf{L}_\beta|D_2 \times S^1\rangle = \delta_{\alpha\beta}$, see (3.22).

In $\underline{3\text{-}24}$, there exists a UV Wilson loop whose half-index is equal to $\chi_{(3,3,1)}^{W_3(3,7)}$ up to a $q$-prefactor,

$$q^2 I_A[W_{(-2,2,1)}] = q^4 I_A[W_{(-4,3,1)}] = q^{-3/28}\chi_{(3,3,1)}^{W_3(3,7)}. \tag{5.8}$$

However, they are not simple:

$$\langle W_{(-2,2,1)}^{-}W_{(-2,2,1)}^{+}\rangle_{\text{sci}} = \langle W_{(-4,3,1)}^{-}W_{(-4,3,1)}^{+}\rangle_{\text{sci}} = 3. \tag{5.9}$$

This suggests that the Wilson loops can be expressed as linear combinations of three IR simple objects. Based on our computations, we propose the following expressions:

$$\begin{aligned}
q^2 W_{(-2,2,1)} &\xrightarrow{\text{RG and A-twisting}} \mathbf{L}_{-\frac{4}{7}} + \mathbf{L}_{-\frac{3}{7}} - \widetilde{\mathbf{L}}_{-\frac{3}{7}} , \\
q^4 W_{(-4,3,1)} &\xrightarrow{\text{RG and A-twisting}} \mathbf{L}_{-\frac{4}{7}} - \mathbf{L}_{-\frac{3}{7}} + \widetilde{\mathbf{L}}_{-\frac{3}{7}} .
\end{aligned} \tag{5.10}$$

Note that there are two primaries with a conformal dimension of $-3/7$ in the $W_3$ minimal model, and the corresponding bulk simple objects are denoted by $\mathbf{L}_{-\frac{3}{7}}$ and $\widetilde{\mathbf{L}}_{-\frac{3}{7}}$. They are related to each other by charge conjugation and they share the same conformal character. The above identification is also consistent with $\langle W_{(-2,2,1)}^{-}W_{(-4,3,1)}^{+}\rangle_{\text{sci}} = -q^{-2}$.

## Acknowledgments

The work of DG is supported in part by the National Research Foundation of Korea grant NRF-2022R1C1C1011979. The work of HK is supported by the Ministry of Education of the Republic of Korea and the National Research Foundation of Korea grant NRF-2023R1A2C1004965. The work of BP is supported by National Research Foundation of Korea grant NRF-2022R1C1C1011979 and NRF-4199990114533. The work of SS is supported by the US Department of Energy under grant DE-SC0010008. We also acknowledge the support by the National Research Foundation of Korea (NRF) Grant No. RS-2024-00405629.

## A  Supersymmetric partition functions

In this appendix we summarize the conventions and formalism used in the present work for computing the supersymmetric partition function $\mathcal{Z}_{\mathcal{M}_{g,p}}$. Here $\mathcal{M}_{g,p}$ is an oriented circle bundle of degree $p \in \mathbb{Z}$ over a closed Riemann surface $\Sigma_g$. For a more in-depth discussion of this topic refer to [59, 60].

### A.1  General structure of partition functions on $\mathcal{M}_{g,p}$

For our purposes the main object to be computed is $\mathcal{Z}_{\mathcal{M}_{0,1}}$, as $\mathcal{M}_{0,1} \simeq S^3$ and thus it is necessary for F-maximization calculations. However, the other "twisted partition functions", $\mathcal{Z}_{\mathcal{M}_{g,p}}$, are useful for checking dualities between theories so the general case will be discussed here.

The partition function $\mathcal{Z}_{\mathcal{M}_{g,p}}$ for an $\mathcal{N} = 2$ SUSY Chern-Simons-Yang-Mills matter theory with gauge group $G$, flavor group $G_F$ and chiral multiplets $\Phi_i$ can be defined using two pieces of data: the *effective twisted superpotential* $\mathcal{W}(u,v)$ and the *effective dilaton* $\Omega(u,v)$. These are functions of $u_a$ and $v_i$, scalar fields valued in the Cartan subalgebras $\mathfrak{h} \subset \mathfrak{g}$ and $\mathfrak{h}_F \subset \mathfrak{g}_F$ respectively. We also define a similar variable for the background R-symmetry vector multiplet which we will denote $v_R$[16]. Further it is useful to define single-valued fugacities for these variables

$$x_a = e^{2\pi i u_a}, \ \ y_i = e^{2\pi i v_i}. \tag{A.1}$$

In what follows, the discussion is not general and only refers to the forms of $\mathcal{W}$ and $\Omega$ which are necessary for the present work.

The *effective twisted superpotential* $\mathcal{W}(u,v)$ is broken down into two pieces, one originating from the Chern-Simons terms of the theory and one originating from the one-loop contribution of the $\Phi_i$[17]:

$$\mathcal{W}(u,v) = \mathcal{W}_{\mathrm{CS}}(u,v) + \mathcal{W}_{\Phi}(u). \tag{A.2}$$

For the $\mathcal{T}[K, \emptyset]$ theory in (3.1), they are explicitly given by

$$\mathcal{W}_{\mathrm{CS}}(u,v) = \frac{1}{2}\sum_{a,b} K_{ab}u_a u_b + \frac{1}{2}(1 + 2v_R)\sum_a K_{aa}u_a + \sum_{a,i}\delta_{ai}u_a v_i, \tag{A.3}$$

$$\mathcal{W}_{\Phi}(u) = \frac{1}{(2\pi i)^2}\sum_a \mathrm{Li}_2\left(x_a\right). \tag{A.4}$$

The *effective dilaton* $\Omega(u)$ for the $\mathcal{T}[K, \emptyset]$ theory

$$\Omega(u) = \frac{1}{2\pi i}\sum_a \ \log\left(1 - x_a\right). \tag{A.5}$$

---

[16]The choice of $v_R$ is not entirely free as will be discussed later in this appendix.

[17]Importantly this is all done in the "$U(1)_{-\frac{1}{2}}$ quantization" which is arises due to the parity anomaly that exists in 3D gauge theories as discussed in [60].

With the twisted effective superpotential and effective dilaton we now define the building blocks of our partition function. These are the *handle gluing operator* $\mathcal{H}(u,v)$, *fibering operator* $\mathcal{F}(u,v)$, and the *flux operators* $\Pi(u,v)$:

$$\mathcal{H}(u,v) = \exp\left(2\pi i \Omega(u,v) \det_{ab}\left(\frac{\partial^2 \mathcal{W}(u,v)}{\partial u_a \partial u_b}\right)\right), \tag{A.6}$$

$$\mathcal{F}(u,v) = \exp\left(2\pi i \left(\mathcal{W}(u,v) - u_a \frac{\partial \mathcal{W}(u,v)}{\partial u_a} - v_i \frac{\partial \mathcal{W}(u,v)}{\partial v_i}\right)\right), \tag{A.7}$$

$$\Pi_a(u,v) = \exp\left(2\pi i \frac{\partial \mathcal{W}(u,v)}{\partial u_a}\right), \quad \Pi_i(u,v) = \exp\left(2\pi i \frac{\partial \mathcal{W}(u,v)}{\partial v_i}\right). \tag{A.8}$$

$\mathcal{H}(u,v)$ and $\mathcal{F}(u,v)$ can be inserted into the partition function on an $S^2 \times S^1$ background to form the partition function on a $\mathcal{M}_{g,p}$, degree $p$ circle bundle over genus $g$ Riemman surface, i.e.

$$\mathcal{Z}_{\mathcal{M}_{g,p}} = \langle \mathcal{H}^g \mathcal{F}^p \rangle_{S_A^2 \times S^1}, \tag{A.9}$$

where the A subscript indicates we are utilizing the A-twist background on $S^2$ as described in [59]. The *flavor flux operator* $\Pi_i(u,v)^{\mathfrak{n}_i}$ can also be inserted to compute the partition function in the presence of a nontrivial flavor flux $\mathfrak{n}_i$ threaded through the base space $\Sigma_g$. Lastly, the *gauge flux operator*, $\Pi_a(u,v)$ is a rational function in the $x_a$ which defines the *Bethe vacua*[18] as solutions to the system of polynomial equations:

$$\mathcal{S}_{BE} = \left\{\hat{u}_a \mid \Pi_a(\hat{u},v) = 1, \forall a = 1,\ldots, r\right\}. \tag{A.10}$$

Combining everything, the partition function with these operator insertions takes the form:

$$\mathcal{Z}_{\mathcal{M}_{g,p}}(v;v_R,\mathfrak{n}_i) = \sum_{\hat{u} \in \mathcal{S}_{BE}(v;v_R)} \mathcal{H}(\hat{u},v)^{g-1}\mathcal{F}(\hat{u},v)^p \prod_i \Pi_i(\hat{u},v)^{\mathfrak{n}_i}. \tag{A.11}$$

## A.2 R-Symmetry Backgrounds and F-Maximization

In order to use the tools described in the previous section we must have a choice of $v_R$. This is related to a choice of background configuration for the $U(1)_R$ gauge field $A^R$ on $\mathcal{M}_{g,p}$. We will now review the possible R-symmetry backgrounds of the 3D A-model as well as their application to F-maximization and twisted index computations.

### A.2.1 $U(1)_R$ Holonomy and Flux in the 3D A-model

The construction of the 3D A-model necessitates a background configuration for the $U(1)_R$ vector multiplet. This configuration can be described by a choice of $v_R$ and $\mathfrak{n}_R$ which are related to the fiber holonomy and flux of the background $U(1)_R$ gauge field $A^R$. Qualitatively these can be thought of as

$$v_R \text{ `` = ''} -\frac{1}{2\pi}\int_{S^1} A^R, \qquad \mathfrak{n}_R \text{ `` = ''} \frac{1}{2\pi}\int_{\Sigma_g} dA^R, \tag{A.12}$$

though a more proper treatment is given in [60]. Now, the 3D A-model background on $\mathcal{M}_{g,p}$ imposes:

$$v_R \in \begin{cases} \frac{1}{2}\mathbb{Z}, & p \text{ even} \\ \mathbb{Z}, & p \text{ odd} \end{cases}, \qquad \mathfrak{n}_R = g - 1 + v_R p. \tag{A.13}$$

---

[18]For non-abelian gauge group $G$ the action of the Weyl group of $G$ on $\mathcal{S}_{BE}$ must be considered.

Given a choice of representative for $v_R$, it is apparent from (A.12) that integer shifts of $v_R$ correspond to large gauge transformations of $A_\mu^R$ and thus describe the same background. Explicitly we have:

$$(v_R, \mathfrak{n}_R) \sim (v_R + 1, \mathfrak{n}_R + p). \tag{A.14}$$

Recall that for a field with R-charge $r$ the Dirac quantization condition is $r\mathfrak{n}_R \in \mathbb{Z}$ and thus the A-model background generically enforces $r \in \mathbb{Z}$. However, examining the conditions (A.13), we see that when $p \neq 0$, $g - 1 = 0 \pmod{p}$, and $v_R$ is integral[19], a specific representative $v_R$ can be chosen such that $\mathfrak{n}_R = 0$ in which case we are free to allow $r \in \mathbb{R}$. This is crucial for F-maximization where the R-charge must be allowed to take on arbitrary real values.

### A.2.2 F-maximization in the 3D A-model

As stated in section 3.1, the superconformal R-charge of the IR SCFT need not be the same as that of the UV theory. Generically the superconformal R-charge can be a mixture of the reference UV R-charge and any available abelian flavor charges, including those that are emergent in the IR. For $\mathcal{T}[K, \emptyset]$ theory, we parameterize this mixing as in (3.6):

$$R_{\vec{\nu}} := R_* + \vec{\mu} \cdot \vec{T}.$$

In a 3D $\mathcal{N} = 2$ QFT, the free energy $F \equiv -\log|\mathcal{Z}_{S^3}|$ is maximized as a function of $\vec{\mu}$ at the superconformal R-symmetry, a principle suitably named *F-Maximization* [44–46]. Our goal is thus to show how $\mathcal{Z}_{S^3}$ as defined in section A.1 depends on $\vec{\mu}$. It will then be possible to utilize F-maximization to compute the superconformal R-charge of the IR theory.

The mixing of $U(1)_R$ with the abelian flavor symmetry appears as a mixing of the R and flavor symmetry current multiplets

$$j_\mu^{(R)} \longrightarrow j_\mu^{(R)} + \vec{\mu} \cdot \vec{j}_\mu^{(F)}. \tag{A.15}$$

Alternatively, via the minimal coupling between current and vector multiplets, this can be equivalently described as a shift in the flavor vector multiplet

$$\vec{\mathcal{V}}^{(F)} \longrightarrow \vec{\mathcal{V}}^{(F)} + \vec{\mu}\, \mathcal{V}^{(R)}. \tag{A.16}$$

It is then clear from their definitions that the variables in the partition function of section A.1 are affected as follows

$$v_i \longrightarrow v_i + \mu_i v_R, \qquad \mathfrak{n}_i \longrightarrow \mathfrak{n}_i + \mu_i \mathfrak{n}_R. \tag{A.17}$$

Using this we see how the functions of section A.1 change: $\mathcal{W}_{\mathrm{CS}}$ gains a term of the form $v_R \vec{u} \cdot \vec{\mu}$ and the flux operators $\Pi_i$ appearing in (A.11) have their exponent shifted by $\mu_i \mathfrak{n}_R$. The latter fact indicates that if we are working in a background with no flavor flux threaded through $\Sigma_g$ then the R/flavor symmetry mixing turns on such a flux. Equivalently, due to the functional forms of $\mathcal{H}$ and $\Pi_i$ given in (A.6) and (A.8), we can think of this as a shift in $\Omega$ given by

$$\Omega \longrightarrow \Omega + \left( \frac{\mathfrak{n}_R}{g-1} \right) \mu_i \frac{\partial \mathcal{W}}{\partial v_i}. \tag{A.18}$$

With the shifted versions of $v_i$ and $\Omega$ we can set $v_i = 0$ and compute $\mathcal{Z}_{S^3}(\mu_i)$ to determine a theories superconformal R-charge. However, as mentioned previously the flavor and R-charges of operators generically must be integral. This implies that the $\mu_i$ are also generically integral. Only when $g - 1 = 0 \bmod p$ can we choose a representative $v_R$ such that $\mathfrak{n}_R = 0$; allowing R-charges

---

[19]One can find conditions for $v_R$ half-integral, namely $g - 1 = \frac{p}{2} \pmod{p}$, but this will not be important here.

and thus the $\mu_\alpha$ to be arbitrary real numbers. For the case of $S^3 \simeq \mathcal{M}_{0,1}$ such a representative of the R-symmetry background exists. Specifically, we must set $(v_R, \mathfrak{n}_R) = (1,0)$ in order to perform F-maximization calculations, otherwise $\mathcal{Z}_{S^3}(\mu_i)$ is ill-defined for arbitrary real $\mu_i$. Comparing our formulas (3.16) and (A.11), we seem to have two very different descriptions of the 3-sphere partition function which we could use for F-maximization. Fortunately, for our purposes they are equivalent

$$|\mathcal{Z}^K_{S^3_{b=1}}(\vec{\mu})| = |\mathcal{Z}_{\mathcal{M}_{0,1}}(v = \vec{\mu}; v_R = 1, \mathfrak{n}_R = 0)|$$

and thus we can choose either representation to compute the superconformal R charge.

In this work, when we deform the UV theory $\mathcal{T}[K, \emptyset]$ by monopole operators to arrive at $\mathcal{T}[K, \{\mathcal{O}_I\}]$, the effect can be understood as follows. Given a relevant, dressed, monopole operator which we can add to the superpotential, we must insist that its R-charge remains 2 after mixing. In most of the cases studied in this paper, the theories with gauge group $U(1)^r$ were deformed by $r-1$ monopoles. This gives $r-1$ constraints on the $r$ variables $\mu_i$ and thus there is a one parameter family of $\mu_i(\tilde{\nu})$ along which we perform F-maximization:

$$\mu_i(\tilde{\nu}) = c_i + \tilde{\nu} a_i.$$

Where $a_i$ will always be the normalized charge vector of the unbroken $U(1)_A$ subgroup and $c_i$ are constants determined by requiring the superpotential have $R = 2$. In this sense determining the value $\vec{\mu}_0$ at which the theory will be superconformal is reduced to a one-dimensional extremization problem on $|\mathcal{Z}_{\mathcal{M}_{0,1}}(\tilde{\nu})|$. If we define the extremizing value to be $\nu_*$, then $\vec{\mu}_0 = \vec{\mu}(\nu_*)$ which is the reference point defining the affine line (3.9).

## A.3 Modular data of A/B twisted TFTs

The partition functions on $\mathcal{M}_{g,p}$ with $p \in 2\mathbb{Z}$ of the $A$ (or $B$) twisted rank-0 SCFT are conjecturally equal to the A-model partition functions using the $U(1)$ R-charge $R_{\nu=-1} = 2J_3^H$ (or $R_{\nu=1} = 2J_3^C$) in (3.9) and (3.12):

$Z[A\text{-twisted theory on } \mathcal{M}_{g,p\in 2\mathbb{Z}}]$

$$= \left( \mathcal{Z}_{\mathcal{M}_{g,p}}(v; v_R, \mathfrak{n}) \Big|_{v=-\frac{\mu}{2}, v_R=-\frac{1}{2}, \mathfrak{n}=\mu((g-1)-\frac{p}{2})} \right) \Bigg|_{\mu=\mu_0-a} \tag{A.19}$$

$$= \left( \sum_{\hat{u} \in \mathcal{S}_{BE}(v; v_R)} \mathcal{H}(\hat{u}, v)^{g-1} \mathcal{F}(\hat{u}, v)^p \prod_i \Pi_i(\hat{u}, v)^{\mathfrak{n}_i} \Bigg|_{v=-\frac{\mu}{2}, v_R=-\frac{1}{2}, \mathfrak{n}=\mu((g-1)-\frac{p}{2})} \right) \Bigg|_{\mu=(\mu_0-a)},$$

up to an overall phase factor. Alternatively, as a finite semisimple 3D TFT, the partition function can be given as

$$Z[A\text{-twisted theory on } \mathcal{M}_{g,p\in 2\mathbb{Z}}] = \sum_\alpha (S_{0\alpha})^{2-2g} (T_{\alpha\alpha})^{-p}, \tag{A.20}$$

using the modular data, $S$ and $T$, of the TFT. Comparing the two expressions, the modular data of the $A$-twisted TFT is[20]

$$S_{0\alpha}^{-2} = \widetilde{\mathcal{H}}(\hat{u}_\alpha, v), \quad (T_{\alpha\alpha})^{-2} = e^{i\pi\delta} \widetilde{\mathcal{F}}^2(\hat{u}_\alpha, v),$$

$$\hat{u}_\alpha \in \mathcal{S}_{BE}(v, v_R) \text{ with } v = -\frac{\mu}{2} = \left( -\frac{\mu_0 - a}{2} \right) \text{ and } v_R = -\frac{1}{2}, \tag{A.21}$$

---

[20]In previous works [30, 32], $\mathcal{H}_{\text{their}} = \widetilde{\mathcal{H}}_{\text{our}}$ and $\mathcal{F}_{\text{their}} = \widetilde{\mathcal{F}}_{\text{our}}$ and thus the relation is simply $S_{0\alpha}^{-2} = \mathcal{H}_{\text{their}}$ and $(T_{\alpha\alpha})^{-2} = \mathcal{F}_{\text{their}}$

with a $\delta \in \mathbb{Q}$.[21] We define

$$\widetilde{\mathcal{H}}(\hat{u}_\alpha, v) := \mathcal{H}(\hat{u}_\alpha, v) \prod_i \Pi_i^{-2v_i}(\hat{u}_\alpha, v),$$

$$\widetilde{\mathcal{F}}(\hat{u}_\alpha, v) := \mathcal{F}(\hat{u}_\alpha, v) \prod_i \Pi_i^{v_i}(\hat{u}_\alpha, v). \tag{A.22}$$

We expect an injective map from $[W_{\vec{Q}_\alpha}] \in W_{\text{sim}}^A / \sim$, where $\sim$ is the IR equivalence in (3.24), to $\hat{u}_\alpha \in \mathcal{S}_{\text{BE}}(v = -\frac{\mu_0 - a}{2}, \nu_R = -\frac{1}{2})$ satisfying

$$\prod \exp(-2\pi i \vec{Q}_\alpha \cdot \hat{u}_\beta) = \left( \frac{S_{\alpha\beta}}{S_{0\beta}} \text{ of A-twisted TFT} \right) . \tag{A.23}$$

Using this relation, one can fix the matrix elements $S_{\alpha\beta}$ [30, 61].

## A.4   Relation to Nahm's conjecture

We can use the information of section A.1 to compute the modular $T$ for the theories discussed throughout this work at mixing parameter $\mu_a$ and relate them to the discussion in section 2. Note that to compute the $T$ matrix for the A-twisted (B-twisted) theory, one must set $\vec{\mu} = \vec{\mu}_0 - \vec{a}$ ($\vec{\mu} = \vec{\mu}_0 + \vec{a}$). The $\widetilde{\mathcal{F}}$ in (A.22) is

$$\mathcal{F}(u, v) \prod_i (\Pi_i(u, v))^{v_i} = \exp\left( \frac{1}{2\pi i} \sum_a \left( \text{Li}_2(x_a) + \frac{1}{2} \sum_b K_{ab} \log(x_a) \log(x_b) \right) \right). \tag{A.24}$$

The Bethe-equation at $(v, v_R) = (-\frac{\mu}{2}, -\frac{1}{2})$ is given as in (2.6) with $\zeta_a = e^{i\pi\mu_a}$; using this, we have

$$\exp\left( \sum_{a,b} K_{ab} \log(x_b) \log(x_a) \right) = \exp\left( \sum_a \log((-1)^{\mu_a}(1 - x_a)) \log(x_a) \right). \tag{A.25}$$

When $v = -\frac{\mu}{2} \in \mathbb{Z}$ we can write (A.24) (via an abuse of notation, ignoring subtleties of the function's domain) in terms of the Rogers dilogarithm (2.2)

$$\widetilde{\mathcal{F}}(\hat{u}, v) = e^{\frac{1}{2\pi i} \sum_a L(x_a)}. \tag{A.26}$$

Further, using (2.6) once more, the modulus of this function will be related to the Bloch-Wigner function (2.4)

$$|\widetilde{\mathcal{F}}(\hat{u}, v)| = \exp\left( \frac{1}{2\pi} \sum_a \text{Im}(\text{Li}_2(x_a)) + \arg(1 - x_a) \log|x_a| \right) = e^{\frac{1}{2\pi} \sum_a D(x_a)}. \tag{A.27}$$

Note, that if any entry in $\mu_a$ is non-even the application of (A.25) will add terms linear in $\log(x_a)$ to $L(x)$ and $D(x)$ appearing in (A.26) and (A.27).

## B   Characters of rational VOAs

**U(1)$_\mathbf{k}$:**   The characters corresponding to pure $U(1)_k$ Chern-Simons theory are $(0 \le \mu < k)$

$$\chi_\mu^{U(1)_k}(q) = \frac{q^{\mu^2/2k - 1/24}}{(q)_\infty} \sum_{m \in \mathbb{Z}} q^{km^2/2 + \mu m} . \tag{B.1}$$

---

[21]The phase factor $e^{i\pi\delta}$ comes from 3-manifold framing choice, background R-symmetry Chern-Simons term, gravitational Chern-Simons term and etc. All of these factors contribute to a rational value of $\delta$.

**Virasoro minmal models:** The characters of Virasoro minimal models $M(P,Q)$ are $(1 \leq r < P, 1 \leq s < Q)$

$$\chi_{(r,s)}^{M(P,Q)} = \frac{q^{h-c/24}}{(q)_\infty} \sum_{n \in \mathbb{Z}} \left( q^{n^2 PQ + n(Qr-Ps)} - q^{(nP+r)(nQ+s)} \right) ,$$

$$h = \frac{(Qr - Ps)^2 - (P-Q)^2}{4PQ} , \quad c = 1 - \frac{6(P-Q)^2}{PQ} . \tag{B.2}$$

**Super-Virasoro minimal models:** The characters of $\mathcal{N} = 1$ super Virasoro minimal model $SM(P,Q)$ are $(1 \leq r < P, 1 \leq s < Q$ with $r - s \in 2\mathbb{Z})$

$$\chi_{(r,s)}^{SM(P,Q)} = q^{h-c/24} \frac{(-q^{1/2}; q)_\infty}{(q)_\infty} \sum_{n \in \mathbb{Z}} \left( q^{(n^2 PQ + n(Qr-Ps))/2} - q^{(nP+r)(nQ+s)/2} \right) ,$$

$$h = \frac{(Qr - Ps)^2 - (P-Q)^2}{8PQ} , \quad c = \frac{3}{2} \left( 1 - \frac{2(P-Q)^2}{PQ} \right) . \tag{B.3}$$

$\mathbf{L_k(\mathfrak{osp}(1|2))}$: The characters of $L_k(\mathfrak{osp}(1|2))$ modules are

$$\chi^{\mathfrak{osp}(1|2)_k} [\mathbf{M}_i] (z;q) = \sum_{n=1}^{k+1} (-1)^{n-1} \chi_{(n,2i+1)}^{M(k+2,2k+3)}(q) \, \chi[\mathcal{L}_{n,0}^{(k)}](z;q), \tag{B.4}$$

where $\chi[\mathcal{L}_{n,0}^{(k)}]$ is the character of affine $su(2)$ algebra at level $k$ [62, 63].

$\mathbf{W_k(k, k+N)}$ **minimal models:** The character of the highest weight module labeled by $(j_0, j_1, \cdots, j_{k-1})$ in the $W_k(k, k+N)$ minimal model is given by [64]

$$\chi_{(j_0,j_1,\cdots,j_{k-1})}^{W_k(k,k+N)} = \left[ \frac{(q^{k+N}; q^{k+N})_\infty}{(q)_\infty} \right]^{k-1} \prod_{a=1}^{k-1} \prod_{b=0}^{k-1} (q^{j_b + j_{b+1} + \cdots + j_{a+b-1}}; q^{k+N})_\infty , \tag{B.5}$$

where we define $j_i = j_{i'}$ if $i \equiv i' \pmod{k}$, and they should satisfy $\sum_{i=0}^{k-1} j_i = k + N$. For example, the character of $W_3(3, 7)$ minimal model is written as

$$\chi_{(j_0,j_1,j_2)}^{W_3(3,7)} = \left[ \frac{(q^7; q^7)_\infty}{(q)_\infty} \right]^2 (q^{j_0}; q^7)_\infty (q^{j_1}; q^7)_\infty (q^{j_2}; q^7)_\infty (q^{j_0+j_1}; q^7)_\infty (q^{j_1+j_2}; q^7)_\infty (q^{j_2+j_0}; q^7)_\infty ,$$

$$\tag{B.6}$$

up to $q$-prefactor.

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
