# Peer review of "Three Dimensional Topological Field Theories and Nahm Sum Formulas"

_SciPost Physics, doi:SciPost Phys. 19, 128 (2025)_

## Round 1 · Referee Report · Anonymous (Referee 1) · 2025-5-19

Report

In this paper the authors explicitly identify a number of 3d N=2 theories, which flow to a unitary TFT or rank zero N=4 superconformal field theories. The half-indices of such theories take form of Nahm sums and are expected to have interesting modular properties and to generalize well known Nahm conjecture. By scanning a large class of theories defined by a specific choice of parameters (such as Chern-Simons couplings) the authors indeed identify interesting classes of theories with such properties. The author’s result, with two exceptions, include theories with half-indices that agree with modular Nahm sums found by Zagier. The results broaden our understanding of 3d N=2 theories and their interrelations with chiral algebras. For this reason, I recommend the paper for publication. In addition, as the paper includes some review of literature and related results, it it worth noting that related class of 3d N=2 theories, whose partition functions take form of Nahm sums, have been analyzed in the context of knots-quivers correspondence (arXiv: 1707.04017 and following works); in particular, modular properties of such theories have been discussed e.g. in arXiv: 2005.13349.

Recommendation

Publish (meets expectations and criteria for this Journal)

  • validity: high
  • significance: good
  • originality: high
  • clarity: high
  • formatting: excellent
  • grammar: excellent

Author:  Heeyeon Kim  on 2025-09-04  [id 5778]

(in reply to Report 1 on 2025-05-19)

We would like to thank the referee for carefully reading our manuscript. We added two references that the referee mentioned in the second version of the manuscript.

---

## Round 1 · Referee Report · Anonymous (Referee 2) · 2025-8-26

Report

The authors explore the rational VOAs which may arise from the boundary of the 3d $\mathcal{N}=2$ Abelian Chern-Simons matter theories with $r$ chiral multiplets. They choose very special types of Dirichlet boundary conditions for vector and chiral multiplets preserving chiral supersymmetry, for which the half-indices take the same form as Nahm sum. They list examples that allow for the expressions in terms of the rational CFT characters as the candidates. The problem is interesting for many experts working on conformal field theories, supersymmetric field theories and other branches of mathematical physics. However, I am confused with several points in the manuscript that I list in the attached. I would like the authors to clarify them by responding to them.

Attachment

Recommendation

Ask for major revision

  • validity: good
  • significance: good
  • originality: good
  • clarity: good
  • formatting: good
  • grammar: good

Author:  Heeyeon Kim  on 2025-09-04  [id 5777]

(in reply to Report 2 on 2025-08-26)

We would like to thank the referee for carefully reading our manuscript. Here’s the reply to the comments in the order in which they appear in the report. All the page/citation/equation numbers below refer to those in the first version of the manuscript.

1) The matrix A depends on r. More precisely, r is chosen to be the rank of A, as stated explicitly below eq.(1.3). With the choice described in eq. (1.5), and together with the supersymmetric Dirichlet boundary condition (see point 2 and 3 below for its precise definition), the matrix A encodes the boundary ’t Hooft anomaly, which is identified with the matrix K appearing later in the manuscript. In the second version of the manuscript, we emphasized this once again in the text below equation (1.5).

2,3) The boundary condition adopted throughout the paper is the supersymmetric Dirichlet boundary condition that preserves 2d N=(0,2) supersymmetry, as discussed in the paper [54]. This uniquely determines the boundary conditions for all elementary fields in the theory (up to the boundary value of the scalar field which does not affect the index calculation). Additional data, in particular the combination ($\mu_0-a$), is further fixed by the F-maximization and by the choice of topological twist, as discussed in page 7. We acknowledge that this point was not stated clearly in the main text. In the revised version of the manuscript, we have included a more self-contained review of the boundary conditions.

4) The logic behind this specialization is as follows.

The bulk abelian gauge theory is expected to flow to a super-conformal fixed point with enhanced supersymmetry (a “rank-zero theory”), a conjecture motivated by F-maximization. At the super-conformal point, the superconformal R-charge is identified with $R_{\mu_0}$ in eq. (3.12).

Since the IR theory enjoys N=4 supersymmetry, it admits two topological twists, that we denote by $\nu=1$ and $\nu=-1$. Eq (3.12) justifies this definition: For $\nu=1$ (resp. -1), the R-symmetry used to define the twisted spin is given by $2J^C$ (respt. $2J_H$), which corresponds to the usual B-twists (respt. A-twist). We emphasize that the specialization $\nu=\pm 1$ is part of the definition to construct the semi-simple TFTs that can support the rational VOAs of interest. While one may also study boundary VAs for the untwisted superconformal theory at $\nu=0$, in general, these do not yield rational VOAs.

Now, consider the supersymmetric Dirichlet boundary conditions in the UV theory (as defined above). This boundary condition flows in the IR to some boundary condition of the superconformal theory, which in turn maps to a boundary condition of the topologically twisted theory. An important and non-trivial consistency check is whether the BRST supercharge of the twisted theory ($Q_A$ or $Q_B$) is compatible with this boundary condition. This is very subtle to check explicitly, especially due to the presence of monopole super potentials, as we discuss briefly on page 11 and in footnote 10 in the first version of the manuscript. A more careful discussion on this subtlety is given in our earlier works [19,21]. In the present work, we do not attempt a detailed analysis of the $Q_{A/B}$-invariance of the boundary condition in the IR topological theory. Instead, we focus on analyzing the expressions (3.27) for the theories $T [K, {O_I }]$ that are expected to flow to rank-zero SCFTs, and propose candidate boundary VOAs supported by the corresponding TFTs when the conditions are satisfied. We indeed find in section 4 that there are examples (see e.g., 3-6, 3-7 and 3-8) where the index identically vanishes, suggesting that the the boundary condition B may not preserve the twisted supercharges. We acknowledge that this subtlety was not explained very clearly in the main text. In the revised version, we have added an extended paragraph that highlights the open issue.

This choice fixes the data of the topological field theory and hence completely determines the boundary vacuum character. Additional insertions of line operators can also be considered, as discussed in section 3.3. In particular, there exists a class of line operators whose insertions shift the q^(m-linear) factors in the Nahm sum formula, which superficially mimics a change of topological twist. However, following the logic I described above, (essentially via the F-maximization) one can clearly distinguish the origin of such contributions.

5) Here $K$ denotes the bare CS level in the so called “$U(1)_{-1/2}$ quantization”. The UV effective Chern Simons level is $K - \frac12 I$, which is 3/2 in the simplest case that the referee mentions. In this particular example, the theory is mirror dual to its orientation reversal, so it coincides with the description involving $U(1)_{-3/2}$. In v2, we added a comment on our convention and also a new footnote comparing the notations with other references.

6) The identification of the Dirichlet half index of the $U(1)_{3/2} + \Phi$ theory was first established in [19] by three of the authors of this manuscript, prior to the work mentioned by the referee. The reasoning behind our specific specialization is the same as explained above in point 4. While the vacuum character of the $\nu=-1$ twisted theory coincides with the index of the $\nu=1$ twist with a line operator inserted (a fact that is closely related to the self-mirror property of the theory, as explained in [21] by the second author of this manuscript), the correct identification of the VOA vacuum character should be made with the index without any line operator insertion. The explicit boundary OPE computations done in [21] support this identification.

---

## Round 2 · Referee Report · Anonymous (Referee 2) · 2025-9-13

Report
1) I am still confused with the following description after eq.(1.5):
In this context, the $r\times r$ matrix $A$ is identified with the mixed Chern-Simons level matrix $K$.'' The matrices $A$ are determined by the boundary conditions. They are intrinsically 2d data as the half-indices can enumerate the local operators at the 2d boundary. Although $A$ might be calledeffective levels'', they are not the Chern-Simons levels since the usual Chern-Simons term is defined on 3d.
2,3) There are several unclear points in the discussion on the F-maximization. In the Gang-Yamazaki paper [59], the superconformal $U(1)$ R-charge of the scalar field is fixed as $1/3$ by the F-maximization. On the other hand, in Table 2, it is stated that the superconformal R-charge of the free chiral theory $\mathcal{T}_{\Delta}$ corresponds to $R_*+\frac12 F$ with $R_*$ $=$ $0$ and $F=1$. Why do you get different R-charges? In fact, the expansions of the superconformal indices in the main text do not seem to be computed with $R_*$ $=$ $0$. For example, for the $U(1)_{3/2}+\Phi$, the expansion in section 4.1.1 begin with $1-q-(\eta+1/\eta)q^{3/2}-2q^2-...$, which agrees with the expansions found in [59] with the superconformal $U(1)$ R-charge of $\Phi$ being $1/3$.
Another concern is about the axial symmetries. In general for each $\mathcal{N}=2$ charged chiral multiplet there will be the axial symmetry factor. If the authors consider the $U(1)^r$ gauge theory with $r$ charged chirals, there will be $U(1)^r$ axial charges. There may be mixing between them with the R-symmetry. How are they dealt with in the F-maximization? The axial symmetry is suddenly discussed in the paragraph \textbf{Toplogical twisting} on page 8 and denoted by $A$. What is the relation between the $\mathcal{N}=2$ axial symmetry factors and $A$?
Besides, I am not sure whether the superconformal R-charge in the presence of the boundary is simply determined via the F-maximization of the 3d bulk theory as discussed. The system is defined on the 2d boundary that may involve different $U(1)$ global symmetries so the determination of the superconformal R-charge will depend on the choice of boundary conditions in general. In the boundary conditions the authors are considering, the deformed Dirichlet boundary condition $D_c$ can be obtained by choosing the Dirichlet boundary condition $D$ first and then introducing the boundary term involving the 2d Fermi multiplet so that the chiral multiplet scalars can take the non-zero constant values. Then one also needs to consider a mixing of the boundary global $U(1)$ symmetries as the broken gauge group with the R-symmetry at the first step. But such mixing depending on the boundary condition is not discussed in the text.
4) I cannot follow the logic. But let us assume that the bulk 3d theories have the enhanced $\mathcal{N}=4$ supersymmetry in the IR. Then if I understand correctly, the R-charge of the scalar field will be fixed as $1/3$ as discussed in the Gang-Yamazaki paper [59]. But the authors seem to choose different R-charges as mentioned above. In fact, the Dirichlet-half-index in eq.(3.35) is computed with R-charge $0$.
5) The CS level in the $U(1)_{-1/2}$ quantization can be used for the bulk theory itself if you want as in the paper [62]. But it looks an abuse of notation in the presence of the boundary. What do you mean by the bare CS levels in the $U(1)_{-1/2}$ quantization in the presence of the boundary? Related to the point 1), the Chern-Simons term is defined on 3d. If the authors would like to call $K$ the CS levels, they should be defined in the 3d bulk which should be intrinsically independent of the UV boundary conditions. If $K$ are defined as the 3d data, then they should not be identified with the matrices $A$.
6) If the identification of the Dirichlet half index of the $U(1)_{3/2}+\Phi$ was already made in [19], it should be better to cite it to make it clear that the identification is not a new result. But neither citations nor comments are added in the modified draft.
There are further comments.
7) On page 10: While the Wilson line is discussed in section 3.3, what symmetry is associated with the charges $\vec{Q}$? When the deformed Dirichlet boundary condition $D_c$ is chosen, the boundary global $U(1)^r$ symmetries corresponding to the gauge group will be broken.
8) On page 11: ...boundary torus. [53-56]'' will be...boundary torus [53-56].''
9) There are many expansions of the superconformal indices in section 4. I could not get the same answers with $R=0$ as discussed in section 3. Also it is briefly stated that some of the indices are the same as some other indices. But are they only checked by looking at the first few terms in the expansions or are they proved analytically?
10) On footnote 18; ``which is arises'' is a typo.
Recommendation
Ask for major revision

Author: Heeyeon Kim on 2025-09-13 [id 5810]
(in reply to Report 1 on 2025-09-13)I would like to thank the referee for the comments.
Regarding comments 1) and 5). As stated in eq (3.4), the UV CS level for the bulk theory is $K-\frac12 I$. The proposal of [56], which we review on page 11-12, is that the matrix $A$ encoding the 2d boundary anomaly is in fact identical to $K$. In the convention of section 3.4.1 of [56], their UV CS level is denoted by (lowercase) $k$ and one finds that
We fully acknowledge that $K$ and $A$ are a priori distinct physical quantities (the former defining the bulk 3d CS level and the latter the boundary anomaly) but the proposal of [56] is precisely that they coincide under the Dirichlet boundary condition. This proposal has been verified in numerous examples, and we adopt it here as our starting point.
Regarding comments 2), 3) and 4). The statements in the references [32, 59] that the superconformal R-charge for the chiral multiplet is 1/3 can be misleading. The superconformal R-charge of a non-gauge-invariant operator cannot be uniquely determined by F-maximization, since it may always be redefined by further mixing with the gauge symmetry. Thus the different R-charge assignment ($R(\Phi) = 0$ in our case versus $R(\Phi) = 1/3$ in [32, 59]) arises simply from a different choice of mixing the R-symmetry with the gauge U(1) symmetry. Concretely,
To be more explicit, let us consider the theory $U(1)_{k = -3/2} + \Phi$, which was studied in [32,59]. In equations (3.7) and (3.8) of our paper, we have
Thus, the superconformal R-charge is
We emphasize again that the R-symmetry of gauge-variant operators (such as the chiral fields in our theories) cannot be uniquely determined by F-maximization, since the R-symmetry can be redefined by mixing with the gauge symmetry. F-maximization is insensitive to such an auxiliary mixing parameter, because we are integrating over the gauge fields in the path integral. At the level of the localization computation on $S^3$, this mixing amounts merely to shifting the dummy integration variable $Z \rightarrow Z + \text{const}$.
When we consider the half-index with the $D_c$ boundary condition on chiral multiplets, the gauge group is broken at the boundary, and we are further required to impose $R(\Phi) = 0$ in order to preserve the $U(1)_R$ symmetry. As explained in [56] (see the paragraph above section 2.1.1 in loc.cit.), giving a charged chiral $D_c$ boundary condition with $R(\Phi) = 0$ is part of the UV definition of the superconformal boundary conditions.
Finally, regarding the referee’s comment “If the authors consider the $U(1)^r$ gauge theory with r charged chirals, there will be $U(1)^r$ axial charges”: We do not understand what the referee means by the “axial charges” in this sentence. The theory of $r$ free chiral multiplets has $U(1)^r$ flavor symmetries, but they are all gauged. Our theory has $U(1)^r$ topological symmetries instead, but they are all explicitly broken except for one residual $U(1)$ by the super-potential deformations involving $r-1$ monopole operators. We call this remaining $U(1)$ symmetry the axial symmetry $A$. We perform the F-maximization with respect to this $A$. We emphasize this point clearly around equation (3.10).
7) There is no extra symmetry associated with $\vec Q$. Even though the deformed Dirichlet boundary condition $D_c$ breaks the boundary $U(1)^r$ symmetry, a Wilson line $W = \exp (i\int A)$ in the bulk still exists and can terminate at the boundary, defining a module of boundary VOA. The fact that the boundary condition does not preserve any global symmetry simply implies that the endpoint of this Wilson line does not transform under any boundary global symmetry.
9) We say clearly in the first paragraph of section 4 that the identification of the indices with known characters are conjectures based on the q-series expansion, except for a few cases with known Nahm sum representations.
We are happy to correct the typos mentioned in 8) and 10) and add another reference to our own work in footnote 12, as referee mentioned in 6).

---

## Round 2 · List of Changes

-
pg2. two more references [17,18] are added in footnote 2.
-
pg2. text below (1.5), "A is identified with..." -> "the $r \times r$ matrix A is identified with..."
-
pg 3. A new equation (3.4), highlighting the relation between K and the UV effective CS level.
-
pg 8. A new paragraph called "Topological twisting" added, which explains the relation between the A/B twisting and the R-symmetry embedding more explicitly.
-
pg9. New equations (3.17) and (3.18) and a new paragraph around them, emphasizing the relation between the specialization $\nu=\pm 1$ and the Hilbert series.
-
pg10. A new paragraph below equation (3.22), again emphasizing the relation between the partition function of TFTs and the specialization of the indices.
-
pg11. New paragraphs added at the beginning of section 3.4, which review our choice of boundary conditions.
-
pg12. All of the text on this page is new, providing a detailed explanation on the specialization $\nu=\pm 1$ at the level of the half-index, including the last paragraph on the compatibility of the boundary condition B with the twisted supercharge, highlighting the open issue in our paper (A brief version of this discussion was originally in a footnote 10 of v1, which we removed in v2.)
-
pg 14. A new footnote 12, which compares the notations with other references.

---

## Round 3 · List of Changes

pg2: Below eq (1.5), we added the phrase “under a particular choice of…”, to emphasize that this identification holds only under the specific choice of boundary conditions discussed in section 3.

pg7: added a new footnote 7 clarifying the comparison of conventions with reference [43], and explaining why the claim in that reference should be interpreted with care.

pg12: added new footnote 11 to explain once again the choice R_*(Phi)=0 in the context of the half-index calculation, as a part of super conformal boundary condition.

pg12: Below (3.35), added the sentence “Notice that…” to emphasize that K is the boundary anomaly coefficient under our convention eq (3.4).

In addition, we corrected several minor typographical errors.

---

## Editorial Decision

published